# Hypoxia-inducible factor-1α is a critical transcription factor for IL-10-producing B cells in autoimmune disease

Xianyi Meng[1], Bettina Grötsch[1], Yubin Luo[1], Karl Xaver Knaup[2], Michael Sean Wiesener[2], Xiao-Xiang Chen[3], Jonathan Jantsch[4], Simon Fillatreau[5], Georg Schett[1] & Aline Bozec[1]

Hypoxia-inducible factors (HIFs) are key elements for controlling immune cell metabolism and functions. While HIFs are known to be involved in T cells and macrophages activation, their functions in B lymphocytes are poorly defined. Here, we show that hypoxia-inducible factor-1α (HIF-1α) contributes to IL-10 production by B cells. HIF-1α regulates IL-10 expression, and HIF-1α-dependent glycolysis facilitates CD1d$^{hi}$CD5$^+$ B cells expansion. Mice with B cell-specific deletion of *Hif1a* have reduced number of IL-10-producing B cells, which result in exacerbated collagen-induced arthritis and experimental autoimmune encephalomyelitis. Wild-type CD1d$^{hi}$CD5$^+$ B cells, but not *Hif1a*-deficient CD1d$^{hi}$CD5$^+$ B cells, protect recipient mice from autoimmune disease, while the protective function of *Hif1a*-deficient CD1d$^{hi}$CD5$^+$ B cells is restored when their defective IL-10 expression is genetically corrected. Taken together, this study demonstrates the key function of the hypoxia-associated transcription factor HIF-1α in driving IL-10 expression in CD1d$^{hi}$CD5$^+$ B cells, and in controlling their protective activity in autoimmune disease.

[1] Department of Internal Medicine 3, Friedrich-Alexander-University Erlangen-Nürnberg (FAU) and Universitätsklinikum Erlangen, 91054 Erlangen, Germany. [2] Department of Internal Medicine 4, Friedrich-Alexander-University Erlangen-Nürnberg (FAU) and Universitätsklinikum Erlangen, 91054 Erlangen, Germany. [3] Department of Rheumatology, Renji Hospital Affiliated to Shanghai Jiao Tong University School of Medicine, 20001 Shanghai, China. [4] Institute of Clinical Microbiology and Hygiena, University Hospital of Regensburg, University of Regensburg, 93053 Regensburg, Germany. [5] Institut Necker-Enfants Malades (INEM), INSERM U1151-CNRS UMR 8253, Université Paris Descartes, Sorbonne Paris Cité, Bâtiment Leriche, 75993 Paris, France. Correspondence and requests for materials should be addressed to A.B. (email: aline.bozec@uk-erlangen.de)

B cells are traditionally known for their effector function involved in antigen presentation and antibody secretion upon their differentiation into plasmablasts and plasma cells conferring humoral immunity[1]. However, increasing attention has been directed to the immune regulatory function of B cells[2,3]. This regulatory function is associated with their production of anti-inflammatory cytokines such as IL-35, TGF-β, and in particular IL-10[4–6]. Previous studies have shown that CD1d[hi]CD5[+] B cells, transitional 2-marginal zone precursors (T2-MZP; CD23[hi]CD21[hi]IgM[+]), antibody-secreting cells (CD44[hi]CD138[+] plasmablasts), and peritoneal CD5[+] B1a cells can, through the production of IL-10, suppress pathogenic T cells and inhibit autoimmune inflammatory diseases such as experimental autoimmune encephalomyelitis (EAE), arthritis, and colitis, as well as contact hypersensitivity[7–10].

Hypoxia-inducible factors (HIFs) are heterodimeric transcription factors, consisting of an oxygen-labile alpha subunit (HIF-α) and a constitutively stable beta subunit (HIF-β), that exert pivotal roles in inducing cellular responses to hypoxia[11]. While hypoxia causes alpha subunits stabilization and induction of respective target genes, HIF-1α and HIF-2α are hydroxylated by prolyl hydroxylases (PHD) and degraded after binding protein von Hippel Lindau (pVHL) under normoxic conditions[12,13]. HIFs were shown to be involved in innate and adaptive immune activation. In macrophages, HIF-1α increases cell motility and the expression of pro-inflammatory cytokines[14,15]. In adaptive immunity, HIF-1α has been shown to promote Th17 cell development and to enhance the expression of cytolytic molecules such as granzyme B and perforin in CD8 T cells[16,17]. The function of HIFs in B cells, however, is incompletely determined. Interestingly, abnormalities of peritoneal B1 cells and high levels of IgG and IgM antibodies directed against dsDNA have been described in *Hif1a*-deficient chimeric mice[18], suggesting a possible regulation of B cell functions by HIF-1α.

In this study, we delineate the function of HIFs in B cells during autoimmune disease with a particular interest in IL-10-producing CD1d[hi]CD5[+] B cells. We generated B cell-specific *Hif1a* or *Hif2a* mutant mice to test the influence of HIFs on B cell cytokine production and on the course of autoimmune disease. B cell activation through B cell antigen receptor (BCR) induces upregulation of HIF-1α expression, and B cell-specific ablation of *Hif1a*, but not *Hif2a*, impairs IL-10 production by B cells. HIF-1α transcriptionally regulates *Il10* gene expression in cooperation with phosphorylated-STAT3, and is required to establish the glycolytic metabolism driving CD1d[hi]CD5[+] B cells expansion. Furthermore, compared with wild-type (WT), mice lacking HIF-1α in B cells have exacerbated collagen-induced arthritis (CIA) and EAE, which can be rescued by ectopic expression of IL-10 in *Hif1a*-deficient CD1d[hi]CD5[+] B cells and their adoptive transfer in vivo. Our findings reveal HIF-1α as a critical transcription factor for IL-10 production by B cells. HIF-1α expression controls CD1d[hi]CD5[+] B cells expansion and may be considered as a potential target in autoimmune disease.

## Results

**HIF-1α expression increases in activated B cells**. To investigate the role of HIFs in B cells, the expression of HIF-1α and HIF-2α was determined in C57BL/6 WT splenic B cells stimulated with lipopolysaccharide (LPS) (10 μg/ml) or anti-IgM (10 μg/ml). Even under normoxic conditions, *Hif1a* mRNA expression is induced in B cells stimulated with LPS or anti-IgM (Fig. 1a), whereas the expression of *Hif2a* is almost undetectable and remains unchanged when analyzed in fold change (Fig. 1a). Accordingly, HIF-2α protein is hardly detectable, whereas HIF-1α protein increases at 4, 8, and 12 h after LPS or anti-IgM

stimulation in B cells (Fig. 1b). Since HIF-1α induction by LPS has been already reported to be dependent on NF-κB signaling[19], we also checked whether this pathway is effective in B cells. Indeed, knockdown of RelA not only decreases p65 phosphorylation but also HIF-1α protein level in B cells stimulated by LPS for 4 h (Supplementary Fig. 1a).

Since B cells stimulation by anti-IgM also induces HIF-1α (Fig. 1b), we delineated the pathways of HIF-1α induction in BCR-stimulated B cells. Therefore, ERK and STAT3 proteins levels were analyzed. As shown in Fig. 1c, phosphorylated-ERK (pERK) and phosphorylated-STAT3 Ser727 (pSTAT3[727]) are increased in splenic B cells after anti-IgM stimulation, whereas phosphorylated-STAT3 Tyr705 (pSTAT3[705]) is virtually undetectable. Using specific inhibitor of ERK, STAT3, and AKT pathways, which are not affecting B cell viability (Supplementary Fig. 1b), we analyzed the pathway essential for HIF-1α protein expression. Indeed, HIF-1α protein induction is suppressed in a dose-dependent manner when BCR-stimulated B cells are treated with ERK or STAT3 inhibitors, but not AKT inhibitor treatment (Fig. 1d). Similarly, decrease of HIF-1α protein is observed when STAT3 or ERK are knocked down in B cells using siRNA approach (Supplementary Fig. 1c). Interestingly, STAT3[727] phosphorylation is decreased after B cell treatment with ERK inhibitor (Fig. 1d), suggesting that phosphorylation of ERK is essential for STAT3[727] phosphorylation.

Next, we determined whether pSTAT3[727] could also transcriptionally regulate *Hif1a* gene expression. To do so, chromatin immunoprecipitation (ChIP) analysis was performed on a putative STAT3 binding site on *Hif1a* promoter at −309 bp/−319 bp from the transcription starting site (TSS) (Fig. 1e). Indeed, low level of pSTAT3[727] can bind to *Hif1a* promoter in splenic B cells in homeostasis (Fig. 1e). Interestingly, pSTAT3[727] binding on *Hif1a* promoter is strikingly enhanced in BCR-mediated activated B cells (Fig. 1e). Our results demonstrate that HIF-1α is increased at mRNA and protein levels in LPS-treated B cells via the NF-κB pathway and in BCR-stimulated B cells via ERK–STAT3 activation.

**B1a population is reduced in *Mb1[cre]Hif1a[f/f]* mice**. To determine the roles of HIF-1α and HIF-2α during B cell development in vivo, we bred mice carrying a loxP-flanked *Hif1a* or *Hif2a* allele with mice expressing cre recombinase from the *Mb1* promoter to delete *Hif1a* or *Hif2a* specifically in B lymphocytes (referred to herein as *Mb1[cre]Hif1a[f/f]* or *Mb1[cre]Hif2a[f/f]* mice). As expected, HIF-1α or HIF-2α protein is completely abolished in splenic B cells but not in T cells isolated from *Mb1[cre]Hif1a[f/f]* or *Mb1[cre]Hif2a[f/f]* mice compared with WT control mice (Supplementary Fig. 1d). Next, flow cytometric analysis of the B cell subpopulations in *Mb1[cre]Hif1a[f/f]*, *Mb1[cre]Hif2a[f/f]*, and WT control mice were performed (Fig. 2a). No difference can be detected in the populations of pre-pro-B, pro-B, pre-B, immature, and recirculating B cells (Hardy fractions A–F) in WT and mutant mice (Fig. 2b). The splenic B cell subpopulations, transitional type 1 and 2 as well as follicular cells, are also similar in *Mb1[cre]Hif1a[f/f]*, *Mb1[cre]Hif2a[f/f]*, and control mice (Fig. 2c, d). However, percentage and absolute numbers of marginal zone B cells are moderately decreased in *Mb1[cre]Hif1a[f/f]* mice compared to WT mice (Fig. 2d). Next, we analyzed peripheral B cell subsets in inguinal lymph nodes and blood from *Mb1[cre]Hif1a[f/f]*, *Mb1[cre]Hif2a[f/f]*, and WT mice. HIF-1α or HIF-2α deletion does not alter the immature and mature B cell populations in the periphery (Fig. 2e, f). Interestingly, only B1a cell number is drastically decreased in the peritoneum of *Mb1[cre]Hif1a[f/f]* mice when compared to WT or *Mb1[cre]Hif2a[f/f]* mice, whereas no difference is observed for B1b cells (Fig. 2g).

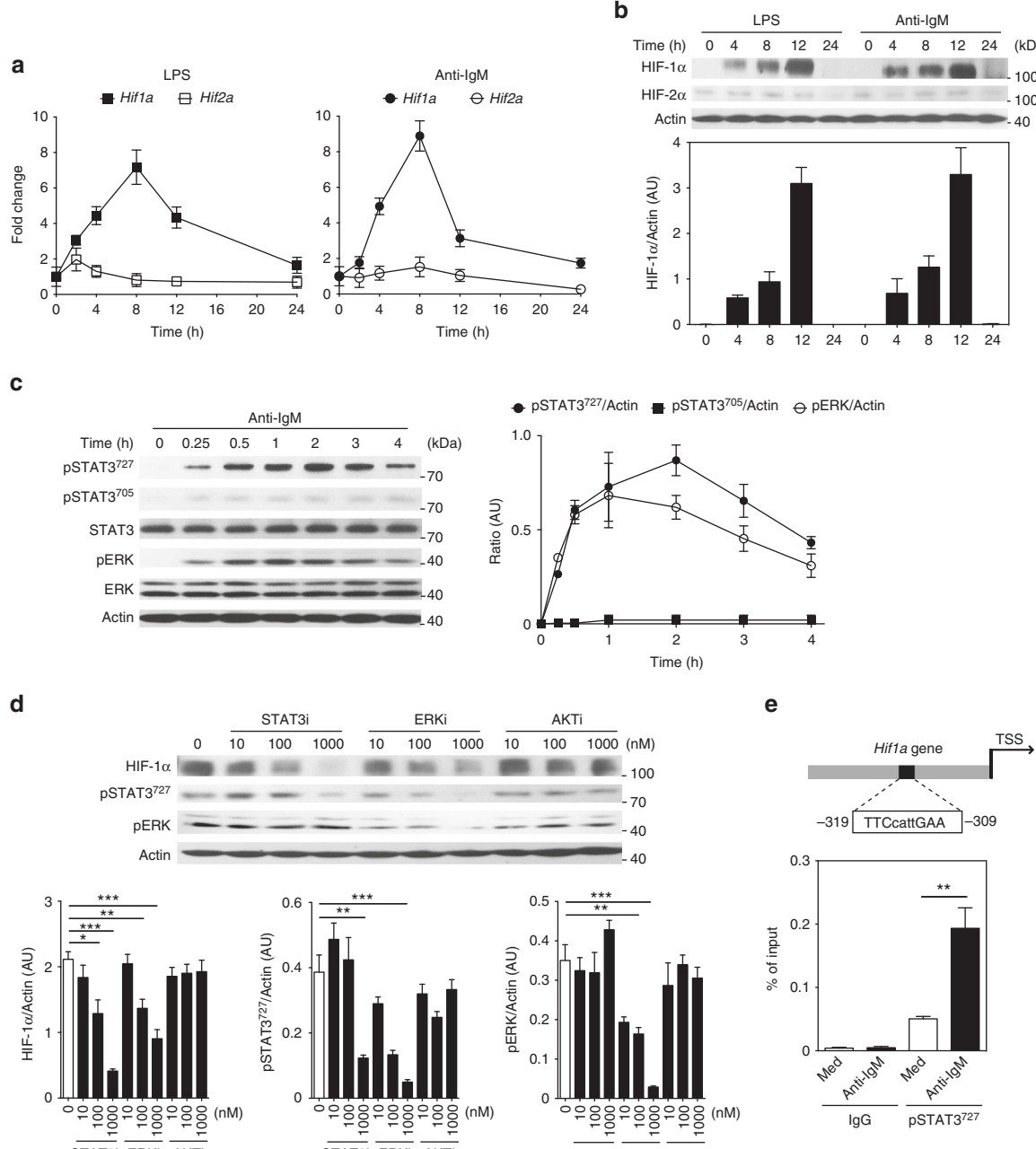

**Fig. 1** Increased HIF-1α expression in activated B cells. **a** Quantitative RT-PCR analyses of *Hif1a* and *Hif2a* in wild-type (WT) splenic B cells stimulated with lipopolysaccharide (LPS) or anti-IgM at indicated time points ($n = 4$ at each time point). Values at 0 h were set as 1. **b** Expression of HIF-1α and HIF-2α in WT splenic B cells stimulated with LPS or anti-IgM at indicated time points ($n = 3$ experiments in duplicate). **c** Western blot and densitometry analysis of phospho-STAT3 Ser727 (pSTAT3[727]), phospho-STAT3 Tyr705 (pSTAT3[705]), total STAT3, phospho-ERK (pERK), and total ERK in WT splenic B cells after stimulation with anti-IgM at indicated time points ($n = 3$ experiments in duplicate). **d** Western blot and densitometry analysis of HIF-1α, pSTAT3[727], and pERK in anti-IgM-stimulated B cells with or without STAT3, ERK, or AKT inhibitors treatments for 4 h ($n = 3$ experiments in duplicate). **e** Scheme of *Hif1a* promoter indicating the potential STAT3 binding site position and enrichment of pSTAT3[727] on *Hif1a* promoter in splenic B cells 4 h after stimulation with anti-IgM or medium (Med) ($n = 4$ for all groups). Data are shown as mean ± s.e.m. Pictures are representative of three (**a**–**d**) or four (**e**) independent experiments. *$P < 0.05$; **$P < 0.01$, and ***$P < 0.001$ (two-tailed unpaired Student's $t$-test) (see also Supplementary Figure 1)

To further determine the effects of HIF-1α and HIF-2α on B cell functions in vitro, proliferation and apoptosis rates were examined in splenic B cell stimulated with LPS, anti-CD40, or anti-IgM. Of note, *Hif1a*- or *Hif2a*-deficient splenic B cells have a normal proliferative or survival ratio after stimulation (Supplementary Fig. 2a, b). We also examined T cell independent (TI) antibody responses and T cell dependent (TD) antibody responses in *Mb1*[cre]*Hif1a*[f/f] and *Mb1*[cre]*Hif2a*[f/f] mice. Antigen-specific antibody production is similar in *Mb1*[cre]*Hif1a*[f/f] or *Mb1*[cre]*Hif2a*[f/f] mice and WT controls, indicating that HIFs are not essential for TI or TD antibody responses (Supplementary Fig. 2c–h). Altogether, these data show that HIF-2α has no essential role during B cell development, whereas HIF-1α is important for the B1a population in the peritoneum.

**HIF-1α deficiency causes CD1d<sup>hi</sup>CD5<sup>+</sup> B cell defects**. Previous studies have shown that B1a cells possess regulatory functions and produce the anti-inflammatory cytokine IL-10 after activation[20,21]. To address whether IL-10 is altered by the loss of HIF-1α or HIF-2α in B cells, IL-10 intracellular staining in B cells was performed. As shown in Fig. 3a, the frequency of IL-10 positive (IL-10<sup>+</sup>) B cells is decreased in bone marrow, spleen, inguinal lymph nodes, and peritoneal cavity of *Mb1<sup>cre</sup>Hif1a<sup>f/f</sup>* mice compared to *Mb1<sup>cre</sup>Hif2a<sup>f/f</sup>* or WT mice. In accordance, HIF-1α intracellular staining in IL-10<sup>+</sup> and IL-10<sup>−</sup> B cells reveal an

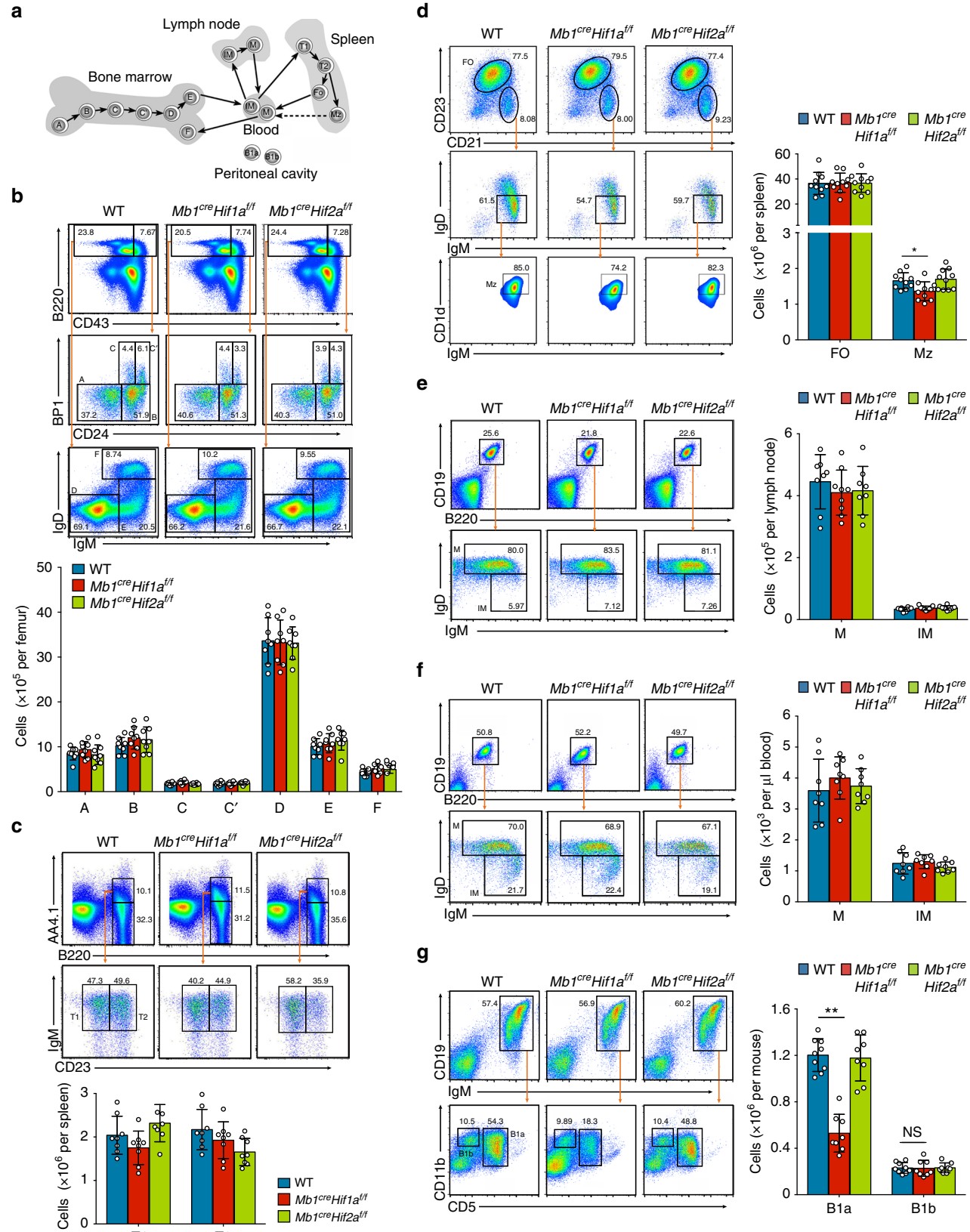

increased level of HIF-1α protein in IL-10$^+$ B cells (Supplementary Fig. 3a). Because IL-10-producing B cells have been described in different B cell subpopulations such as CD1d$^{hi}$CD5$^+$CD19$^+$ B cells[22] and CD23$^+$CD21$^{hi}$IgM$^+$(T2-MZP) B cells[23], we speculated that these two subsets are modified in $Mb1^{cre}Hif1a^{f/f}$ mice in vivo. It is noteworthy that percentage and absolute numbers of CD1d$^{hi}$CD5$^+$ or T2-MZP B cells are reduced in $Mb1^{cre}Hif1a^{f/f}$ mice compared to WT littermates (Fig. 3b and Supplementary Fig. 3b). Less BrdU-positive cells are observed in CD1d$^{hi}$CD5$^+$ and T2-MZP populations of $Mb1^{cre}Hif1a^{f/f}$ mice compared to WT control mice (Fig. 3c and Supplementary Fig. 3c), implying a defect of regulatory B cell proliferation in $Hif1a$-deficient mice. In accordance, IL-10 intracellular staining in CD1d$^{hi}$CD5$^+$ B cells confirms the decreased frequency of IL-10$^+$CD1d$^{hi}$CD5$^+$ B cells in $Mb1^{cre}Hif1a^{f/f}$ mice (Fig. 3d). Moreover, analysis of anti-inflammatory cytokines expression in sorted CD1d$^{hi}$CD5$^+$ B cells from $Mb1^{cre}Hif1a^{f/f}$ mice reveals a significant decrease in $Il10$ mRNA expression as well as a decreased IL-10 production after stimulation (*$P < 0.05$ and **$P < 0.01$, by t-test; Fig. 3e and Supplementary Fig. 3d), whereas $Tgfb$, $P35$, and $Ebi3$ mRNA levels are not altered (Supplementary Fig. 3d). Altogether, these data suggest that HIF-1α is an important factor for the expansion of CD1d$^{hi}$CD5$^+$ B cells, and their IL-10 production.

**HIF-1α regulates glycolysis in CD1d$^{hi}$CD5$^+$ B cells.** Since CD1d$^{hi}$CD5$^+$ B cell number is normal in $Il10$-deficient mice[24], we hypothesized that the reduced IL-10 level is likely not responsible for the reduced CD1d$^{hi}$CD5$^+$ B cell number in $Mb1^{cre}Hif1a^{f/f}$ mice. HIF-1α was previously identified as a key factor for glycolytic activity and glucose metabolism in immune cell function and proliferation[25–27]. To further dissect the expansion of CD1d$^{hi}$CD5$^+$ B cells, we examined the level of HIF-1α in this population. Indeed, HIF-1α protein level is higher in CD1d$^{hi}$CD5$^+$ B cells than in CD1d$^{lo}$CD5$^-$ B cells (Supplementary Fig. 4a). Next, glucose uptake was examined in FACS-sorted CD1d$^{lo}$CD5$^-$ B and CD1d$^{hi}$CD5$^+$ B cells from WT mice. CD1d$^{hi}$CD5$^+$ B cells display a two-fold increase in glucose transport activity compared to CD1d$^{lo}$CD5$^-$ B cells (Fig. 4a), suggesting that CD1d$^{hi}$CD5$^+$ B cells preferentially use glucose metabolism. Moreover, CD1d$^{hi}$CD5$^+$ B cells from $Mb1^{cre}Hif1a^{f/f}$ mice exhibit a lower level of glucose uptake and lactate secretion (Fig. 4b, c) compared to CD1d$^{hi}$CD5$^+$ B cells from WT mice, whereas there is no difference in glucose uptake between $Hif1a$-deficient and WT CD1d$^{lo}$CD5$^-$ B cells (Supplementary Fig. 4b). Accordingly, mRNAs expression of HIF-1α-targeted glycolytic genes, glucose transporter 1 ($Glut1$), pyruvate kinase M2 ($Pkm2$), hexokinase 2 ($Hk2$), lactate dehydrogenase A ($Ldha$),

phosphoinositide-dependent kinase 1 ($Pdk1$), and glucose-6-phosphate isomerase 1 ($Gpi1$), are markedly decreased in $Hif1a$-deficient CD1d$^{hi}$CD5$^+$ B cells compared to WT CD1d$^{hi}$CD5$^+$ B cells (Fig. 4d). Next, we delineated whether the high glycolytic activity of CD1d$^{hi}$CD5$^+$ B cells was critical for the expansion of CD1d$^{hi}$CD5$^+$ B cells. As shown in Fig. 4e, partial inhibition of glycolysis by treatment with competitive glycolytic inhibitor 2-deoxyglucose is sufficient to inhibit WT CD1d$^{hi}$CD5$^+$ B cells proliferation to a similar level as found in untreated $Hif1a$-deficient CD1d$^{hi}$CD5$^+$ B cells. Taken together, these data suggest that HIF-1α expression controls the expansion of CD1d$^{hi}$CD5$^+$ B cells by orchestrating their high glycolytic activity.

**HIF-1α and STAT3 cooperatively regulate $Il10$ transcription.** To delineate how HIF-1α can regulate IL-10 expression in B cells, splenic B cells from $Mb1^{cre}Hif1a^{f/f}$ and WT mice were cultured under normoxic or hypoxic condition. Interestingly, $Il10$ mRNA expression is strongly increased in B cells cultured under hypoxia compared to normoxia (Fig. 5a). Consistent with the reduced IL-10 production in $Hif1a$-deficient B cells (Fig. 3), $Il10$ mRNA expression is also lower in $Hif1a$-deficient B cells than WT B cells under hypoxic condition (Fig. 5a). We next examined whether $Il10$ gene expression could be transcriptionally regulated by HIFs in B cells. Bio-informatics promoter analysis, using JASPA with the consensus core (A/GCGTG), reveals several putative hypoxia-responsive element (HRE) regions (I–V) on $Il10$ promoter (Fig. 5b and Supplementary Fig. 5a). By ChIP assay, we show that HIF-1α can bind to HRE I and HRE II regions under hypoxic condition (Fig. 5c). Interestingly, the pattern of HIF-1α binding is similar to that of histone H3 (trimethylK4) antibodies, whereas no specific binding is detected when using control IgG antibodies (Fig. 5d and Supplementary Fig. 5b), suggesting that these regions are transcriptionally active under hypoxia. Next, luciferase reporter assays with putative HRE constructs were performed in 293T cells after hypoxic or normoxic culture. As expected, the luciferase activity of the HRE I and HRE II constructs are increased under hypoxic condition, suggesting that HIF-1α activates $Il10$ transcription through HRE I and HRE II regions (Fig. 5e).

Since STAT3 and HIF-1α were previously shown to cooperate on HIF target genes such as $CA9$ and $PGK1$[28], we hypothesized that the highly expressed pSTAT3$^{727}$ in BCR-activated B cells (Fig. 1c) might form a complex with HIF-1α to activate $Il10$ transcription. To test this hypothesis, we confirmed the binding of HIF-1α in B cells after anti-IgM stimulation (Fig. 5f). In addition, HIF-1β can also bind to the HRE I and HRE II regions on the $Il10$ promoter in B cell after anti-IgM stimulation, whereas no binding of HIF-2α or control IgG is detected (Supplementary

**Fig. 2** B1a cell number is reduced in the peritoneal cavity of Mb1creHif1af/f mice. **a** Scheme of developmental, maturation, and migration stages of B cells in bone marrow, spleen, peritoneal cavity, lymph node, and blood. Arrows indicate most likely developmental pathway and dotted arrows indicate still debated pathway. **b** Representative plots and absolute numbers of B-cell subpopulations in bone marrow from $Mb1^{cre}Hif1a^{f/f}$ ($n=8$), $Mb1^{cre}Hif2a^{f/f}$ ($n=8$), and WT control ($n=8$)(cre-negative floxed) mice. The subpopulations analyzed were separated into six populations (fractions A–F) according to the Hardy classification, A: pre-pro-B (B220$^+$CD43$^+$BP-1$^-$CD24$^-$); B: pro-B (B220$^+$CD43$^+$BP-1$^-$CD24$^+$); C: late pro-B (B220$^+$CD43$^+$BP-1$^+$CD24$^{med}$); C': early pre-B (B220$^+$CD43$^+$BP-1$^+$CD24$^{hi}$); D: late pre-B (B220$^+$CD43$^-$IgD$^-$IgM$^-$); E: immature B (B220$^+$CD43$^-$IgD$^-$IgM$^+$); and F: recirculating B (B220$^+$CD43$^-$IgD$^+$IgM$^+$). **c** Representative plots and absolute numbers of transitional type 1 and transitional type 2 B cells in spleen from $Mb1^{cre}Hif1a^{f/f}$ ($n=8$), $Mb1^{cre}Hif2a^{f/f}$ ($n=8$), and WT mice ($n=8$). T1: transitional type 1 B cells (B220$^+$AA4.1$^+$IgM$^+$CD23$^-$); T2: transitional type 2 B cells (B220$^+$AA4.1$^+$IgM$^+$CD23$^+$). **d** Representative plots and absolute numbers of follicular B cells and marginal zone B cells in spleen from $Mb1^{cre}Hif1a^{f/f}$ ($n=10$), $Mb1^{cre}Hif2a^{f/f}$ ($n=10$), and WT mice ($n=10$). FO: follicular B cells (B220$^+$AA4.1$^-$CD23$^+$CD21$^+$); Mz: marginal zone B cells (B220$^+$AA4.1$^-$CD23$^-$CD21$^+$IgM$^+$IgD$^-$CD1d$^{hi}$). **e, f** Representative plots and absolute numbers of B-cell subpopulations in lymph node (**e**) or blood (**f**) from $Mb1^{cre}Hif1a^{f/f}$ ($n=8$), $Mb1^{cre}Hif2a^{f/f}$ ($n=8$), and WT mice ($n=8$). M: mature B (B220$^+$CD19$^+$IgD$^+$IgM$^-$); IM: immature B (B220$^+$CD19$^+$IgD$^+$IgM$^+$). **g** Representative plots and absolute numbers of B1a (CD19$^+$IgM$^+$CD11b$^+$CD5$^+$) and B1b (CD19$^+$IgM$^+$CD11b$^+$CD5$^-$) cells in peritoneal cavity from $Mb1^{cre}Hif1a^{f/f}$ ($n=8$), $Mb1^{cre}Hif2a^{f/f}$ ($n=8$), and WT mice ($n=9$). Data are shown as mean ± s.e.m. Pictures are representative of three independent experiments. NS not significant; *$P < 0.05$ and **$P < 0.01$ (two-tailed unpaired Student's t-test) (see also Supplementary Figure 2)

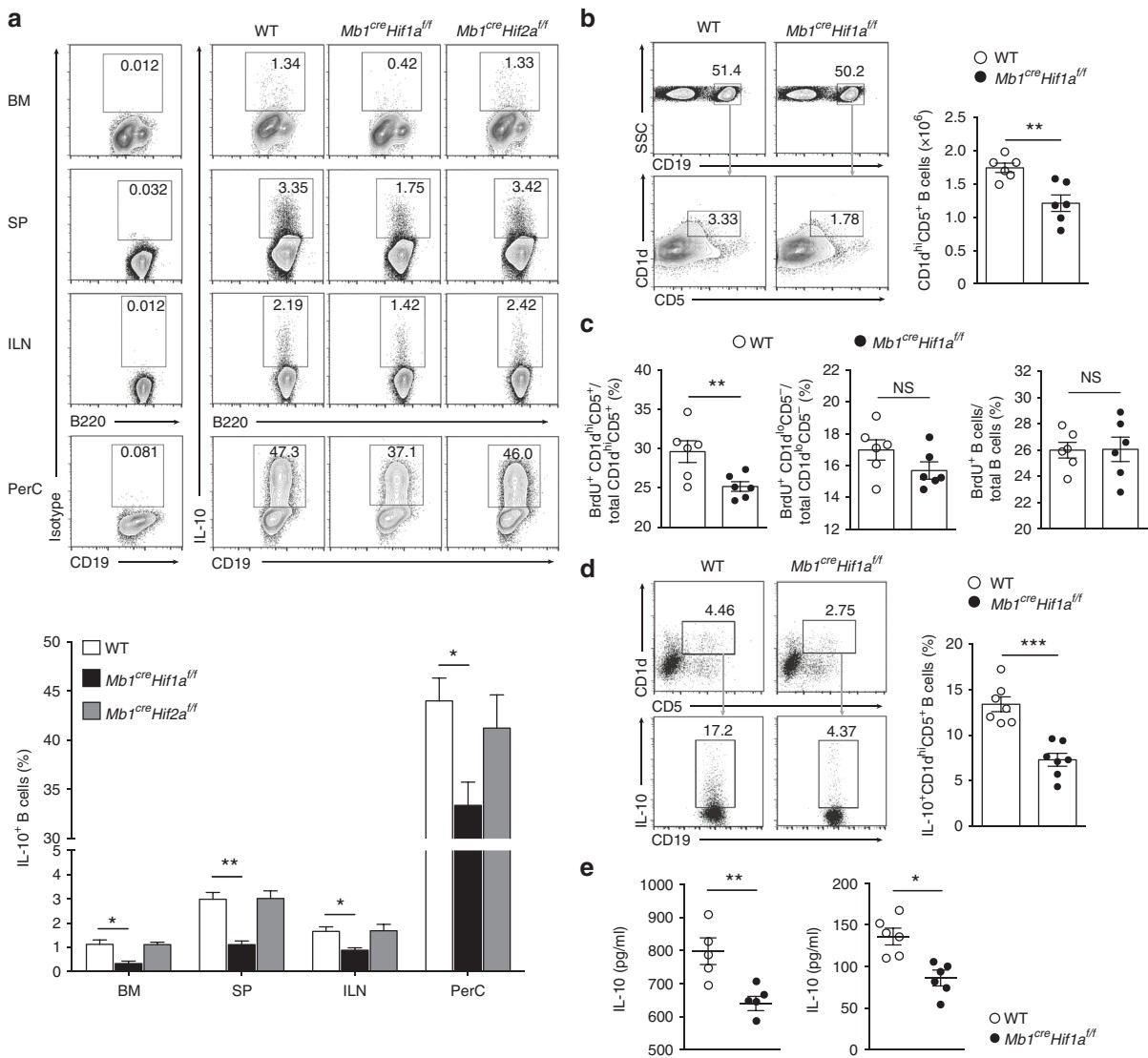

**Fig. 3** HIF-1α deficiency impairs CD1d$^{hi}$CD5$^+$ B cells IL-10 production and expansion. **a** Representative plots and quantification of IL-10-producing B cells in bone marrow (BM), spleen (SP), inguinal lymph nodes (ILN), and peritoneal cavity (PerC) from *Mb1$^{cre}$Hif1a$^{f/f}$*, *Mb1$^{cre}$Hif2a$^{f/f}$*, and control mice ($n = 3$ per group). **b** Representative plots and absolute numbers of CD19$^+$B220$^+$CD1d$^{hi}$CD5$^+$ B cells in spleen from *Mb1$^{cre}$Hif1a$^{f/f}$* ($n = 6$) and WT mice ($n = 6$). **c** Percentage of BrdU$^+$ cells in CD1d$^{hi}$CD5$^+$, CD1d$^{lo}$CD5$^-$, or total splenic B cells isolated from WT ($n = 6$) and *Mb1$^{cre}$Hif1a$^{f/f}$* mice ($n = 6$) 7 days after BrdU treatment. **d** Representative plots and quantification of IL-10$^+$CD1d$^{hi}$CD5$^+$ B cells in spleen from *Mb1$^{cre}$Hif1a$^{f/f}$* ($n = 7$) and WT mice ($n = 7$). **e** IL-10 production by sorted CD1d$^{hi}$CD5$^+$ B cells from *Mb1$^{cre}$Hif1a$^{f/f}$* ($n = 5$, 6) and WT mice ($n = 5$, 6) after stimulation with LPS (left) or anti-IgM (right) for 48 h. Data are shown as mean ± s.e.m. Pictures are representative of three independent experiments. NS not significant; *$P < 0.05$, **$P < 0.01$, and ***$P < 0.001$ (two-tailed unpaired Student's *t*-test) (see also Supplementary Figure 3)

Fig. 5c–e). Next, co-immunoprecipitation of pSTAT3[727] and HIF-1α was performed. Indeed, pSTAT3[727] protein binds to HIF-1α protein in BCR-stimulated B cells (Fig. 5g). Furthermore, two-step ChIP assays pulling-down HIF-1α and pSTAT3[727] sequentially show that pSTAT3[727] could also bind to the HRE I and HRE II regions on *Il10* promoter (Fig. 5h), implying that a complex comprising HIF-1α and pSTAT3[727] might be involved in IL-10 production by BCR-stimulated B cells.

**HIF-1α deficiency in B cells exacerbates autoimmune diseases.** IL-10 production by B cells was previously shown to influence the course of inflammatory autoimmune diseases[6]. Therefore, we hypothesized that HIF-1α in B cells represented a critical node for the modulation of autoimmune diseases. To test this hypothesis, *Mb1$^{cre}$Hif1a$^{f/f}$* mice were subjected to CIA, a standard murine model of arthritis resembling human rheumatoid arthritis. As

shown in Fig. 6a, *Mb1$^{cre}$Hif1a$^{f/f}$* mice show a significantly increased incidence of arthritis after immunization with collagen II (CII) compared to littermate controls (*$P < 0.05$, by Kaplan–Meier analysis with log–rank test). The induction of arthritis is dependent on CII immunization, since no clinical symptom is observed in *Mb1$^{cre}$Hif1a$^{f/f}$* mice without immunization (Fig. 6a, b). Moreover, *Mb1$^{cre}$Hif1a$^{f/f}$* mice exhibit an earlier disease onset and develop higher clinical arthritis scores than WT mice after CII immunization (Fig. 6b). Accordingly, *Mb1$^{cre}$Hif1a$^{f/f}$* mice have an increased paw thickness, synovial inflammation, bone erosions, and number of osteoclasts, confirming an exacerbation of arthritis symptoms in mutant mice (Fig. 6c, d). Similar to TD or TI antibody responses (Supplementary Fig. 2), levels of IgG, IgG1, IgG2a, and IgG2b are not changed in *Mb1$^{cre}$Hif1a$^{f/f}$* and WT arthritic mice (Supplementary Fig. 6a). Next, cytokines mRNA expression pattern was analyzed in synovial tissues. As

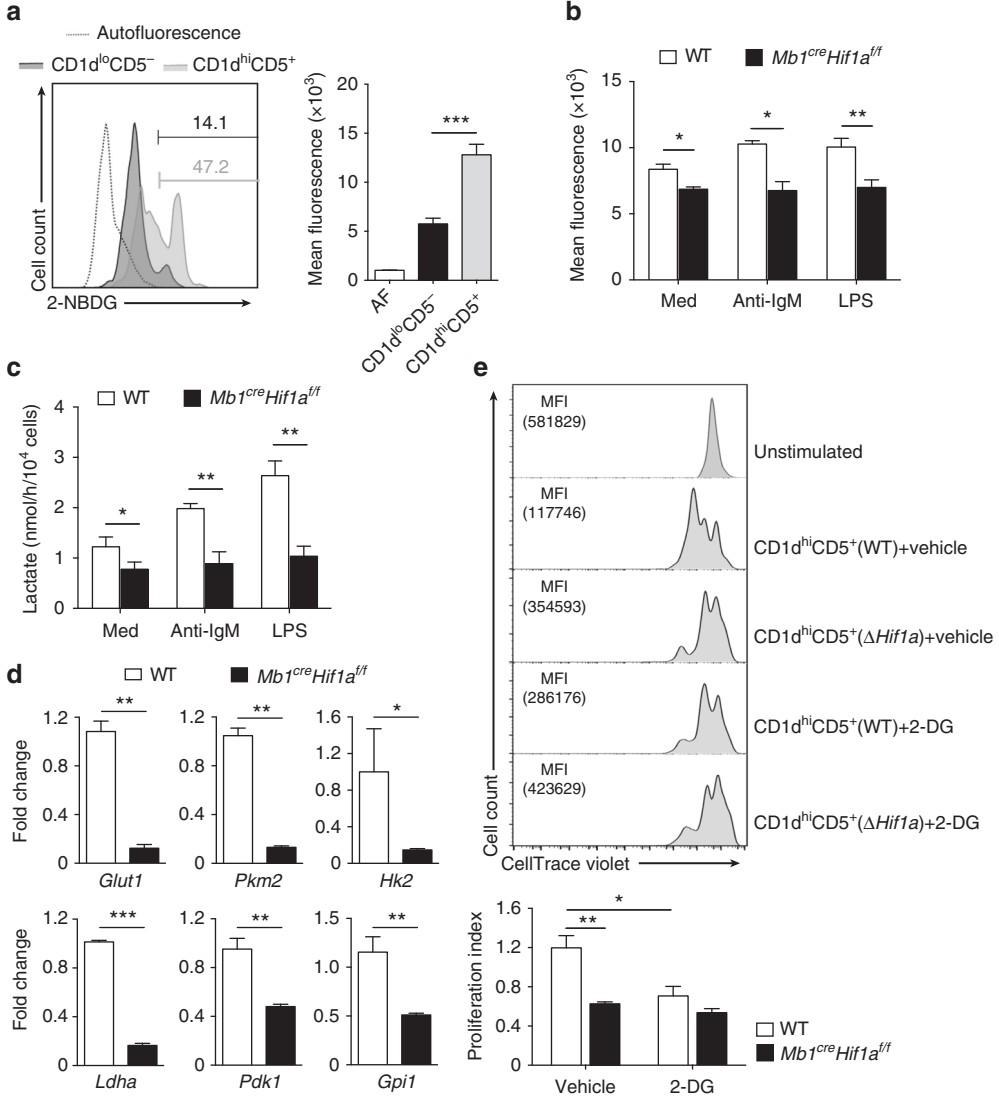

**Fig. 4** HIF-1α-dependent glycolysis is required for CD1d$^{hi}$CD5$^+$ B cells expansion. **a** Glucose transport activity of sorted CD1d$^{lo}$CD5$^-$ and CD1d$^{hi}$CD5$^+$ B cells from WT mice measured by flow cytometry after culturing with fluorescent glucose analog 2-NBDG ($n = 4$ per group). **b** Glucose transport activity of sorted CD1d$^{hi}$CD5$^+$ B cells from $Mb1^{cre}Hif1a^{f/f}$ and WT littermates with medium (Med) or stimulated with anti-IgM or LPS for 6 h ($n = 4$ per group). **c** Lactate secretion by sorted CD1d$^{hi}$CD5$^+$ B cells from $Mb1^{cre}Hif1a^{f/f}$ and WT mice with medium (Med) or stimulated with anti-IgM or LPS for 6 h ($n = 3$ per group). **d** Quantitative RT-PCR analyses of HIF-1α targeted glycolytic genes *Glut*, *Pkm2*, *Hk2*, *Ldha*, *Pdk1*, and *Gpi1* mRNA expression in CD1d$^{hi}$CD5$^+$ B cells from $Mb1^{cre}Hif1a^{f/f}$ and WT mice ($n = 3$ per group). **e** Representative histogram plots and statistical results of proliferation index on CellTrace Violet labeled CD1d$^{hi}$CD5$^+$ B cells from $Mb1^{cre}Hif1a^{f/f}$ and WT littermates treated with vehicle or 0.5 mM 2-deoxyglucose (2-DG) after stimulation with anti-IgM and anti-CD40 for 72 h ($n = 3$ per group). MFI mean fluorescence intensity. Data are shown as mean ± s.e.m. Pictures are representative of four (**a**–**c**) or three (**d**, **e**) independent experiments. *$P < 0.05$, **$P < 0.01$, and ***$P < 0.001$ (two-tailed unpaired Student's *t*-test) (see also Supplementary Figure 4)

expected, an increased level of pro-inflammatory cytokines such as *Tnf*, *Ifng*, *Il17*, *Il1b* mRNA and a reduced level of *Il10* mRNA are detected in synovial tissue of $Mb1^{cre}Hif1a^{f/f}$ mice compared to WT mice after CIA induction (Fig. 6e). Furthermore, the levels of pro-inflammatory cytokines like IL-17 and IFN-γ are higher in the supernatant of CII-stimulated splenocytes from $Mb1^{cre}Hif1a^{f/f}$ mice than WT mice (Fig. 6f). Whereas no change in TGF-β is detected, the level of IL-10 is reduced in CII-stimulated splenocytes and splenic B cells from $Mb1^{cre}Hif1a^{f/f}$ mice (Fig. 6f, g).

To determine whether immune cell populations were altered, Th1 (IFN-γ$^+$CD4$^+$) cells, Th17 (IL-17$^+$CD4$^+$, IL-23R$^+$IL-17$^+$, or GM-CSF$^+$IL-17$^+$) cells, Treg (CD25$^+$Foxp3$^+$CD4$^+$) cells, type 1 regulatory T cells (Tr1) (IL-10$^+$CD4$^+$) cells as well as IL-10$^+$ B cells, ICAM$^+$ B cells, CD73$^+$ B cells, GITRL$^+$ B cells, FasL$^+$ B cells, and PD-L1$^+$ B cells were quantified in the spleen and draining

lymph nodes (dLNs) of $Mb1^{cre}Hif1a^{f/f}$ and WT mice after CII immunization. Regarding the B cell subsets analyses, there is no difference in ICAM$^+$, CD73$^+$, GITRL$^+$, FasL$^+$, or PD-L1$^+$ B cells in spleen and dLNs (Supplementary Fig. 6b, c), only IL-10$^+$ B cells are reduced in $Mb1^{cre}Hif1a^{f/f}$ mice when compared to WT mice (Fig. 6h, i). As expected an increased Th1, Th17 cells, and decreased Tr1 cells are observed in $Mb1^{cre}Hif1a^{f/f}$ mice after CII immunization compared to littermate control mice (Fig. 6h, i). The analyses of pathogenic Th17 subsets reveal that both IL-23R$^+$IL-17$^+$ and GM-CSF$^+$IL-17$^+$ Th17 cells are increased in $Mb1^{cre}Hif1a^{f/f}$ mice after CII immunization compared to littermate control mice (Fig. 6j). Surprisingly, no difference can be detected for the CD25$^+$Foxp3$^+$ Treg population (Fig. 6h, i). In accordance, the suppressive function of Treg cells from $Mb1^{cre}Hif1a^{f/f}$ or WT mice is similar when co-cultured with

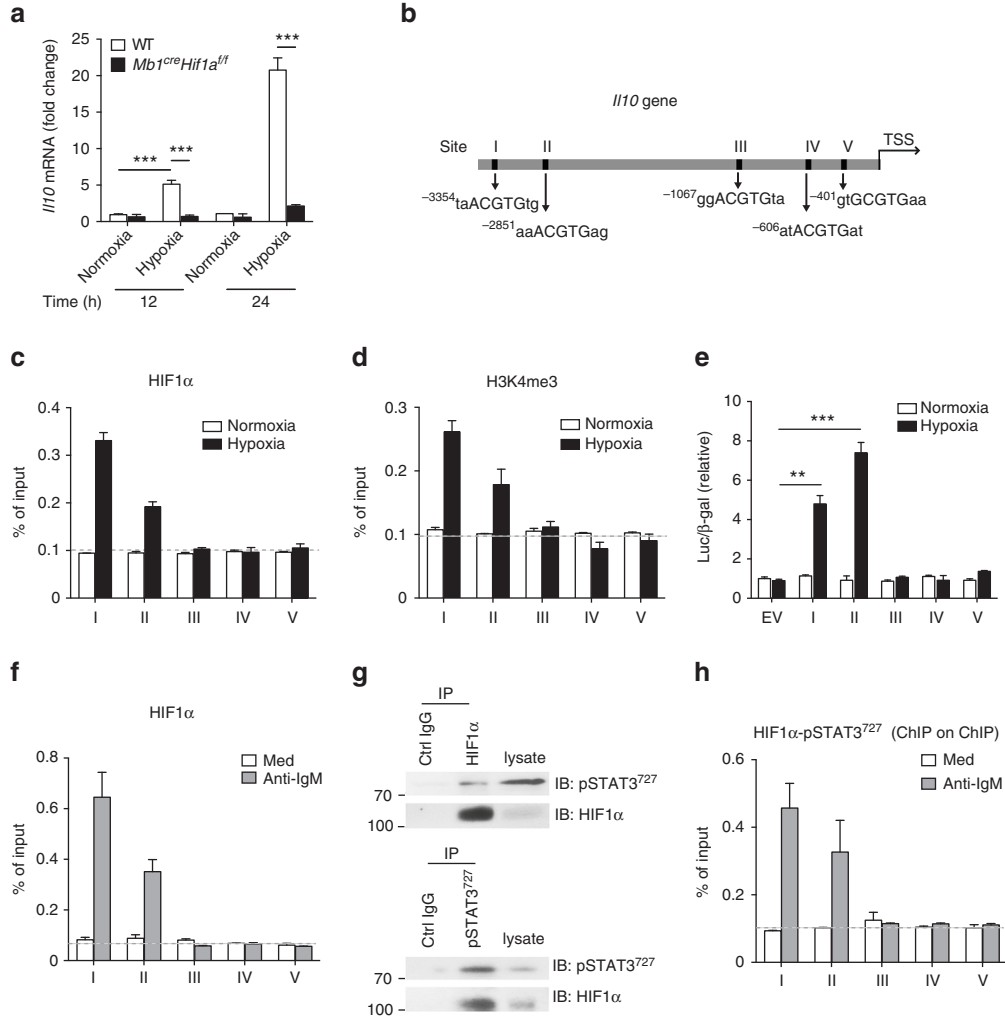

**Fig. 5** HIF-1α and STAT3 cooperatively regulate *Il10* transcription. **a** Quantitative RT-PCR analysis of *Il10* mRNA expression in WT splenic B cells cultured under normoxia or hypoxia for indicated time ($n = 5$ per group). **b** Scheme of *Il10* promoter indicating the predicted HRE regions (I, II, III, IV, and V). Enrichment of HIF-1α (**c**) and H3K4me3 (**d**) in HRE regions of *Il10* promoter in B cells under normoxia or hypoxia for 24 h ($n = 3$ per group). **e** Luciferase activity in 293T cells transfected with pGL3 empty vector (EV), HRE constructs (I, II, III, IV, and V) under normoxic or hypoxic conditions for 24 h ($n = 3$ per group). The Luc/β-gal ratio was normalized to EV at 20% $O_2$. **f** ChIP assays in enriched splenic B cells showing the recruitment of the endogenous HIF-1α on the HRE regions of *Il10* promoter after anti-IgM stimulation for 8 h ($n = 3$ per group). **g** Co-immunoprecipitation of HIF-1α and pSTAT3[727] in enriched splenic B cells stimulated with anti-IgM for 8 h. The whole cell lysates were immunoprecipitated with either anti-HIF-1α or anti-pSTAT3[727] antibodies. **h** ChIP on ChIP assays in enriched splenic B cells pulling down sequentially HIF-1α and pSTAT3[727] showing the HIF-1α-pSTAT3[727] binding on the HRE regions of *Il10* promoter after stimulation with anti-IgM for 8 h ($n = 3$ per group). Data are shown as mean ± s.e.m. Pictures are representative of four (**a**) or three (**c**–**h**) independent experiments. **$P < 0.01$ and ***$P < 0.001$ (two-tailed unpaired Student's *t*-test) (see also Supplementary Figure 5)

effector T cells in vitro (Supplementary Fig. 6d). However, stainings of IL-10 and Foxp3 in CD4⁺ T cells show no difference in Foxp3⁺ Treg cells but a reduced number of IL-10⁺CD4⁺ Tr1 or IL-10⁺Foxp3⁺ Treg cells in spleen and dLNs from *Mb1cre Hif1af/f* mice (Fig. 6h, i and Supplementary Fig. 6e). Altogether, these data indicate that loss of HIF-1α in B cells exacerbates arthritis development likely by impairing the IL-10 production by B cells.

To further confirm the physiological role of HIF-1α in B cells mediated by IL-10 production, the myelin oligodendrocyte glycoprotein peptide (MOG$_{35-55}$) induced EAE was applied to *Mb1cre Hif1af/f* mice and WT littermates. After immunization with MOG$_{35-55}$, *Mb1cre Hif1af/f* mice develop higher clinical scores than WT mice (Fig. 7a). Histopathological analyses reveal an increased number of inflammatory foci and demyelination areas in the spinal cords of *Mb1cre Hif1af/f* mice (Fig. 7b). In addition, increased numbers of infiltrated CD4⁺, CD8⁺ T cells, and F4/80⁺

macrophages are observed in the central nervous system (CNS) of *Mb1cre Hif1af/f* mice compared to WT littermates (Fig. 7c). Like for the CIA model, EAE pathogenesis is associated to Th17, Th1 cells, and production of pro-inflammatory cytokines like IL-17 and IFN-γ. At the peak of EAE disease, increased levels of IL-17 and IFN-γ and a reduced level of IL-10 are found in serum from *Mb1cre Hif1af/f* mice compared to WT littermates (Supplementary Fig. 7a). Furthermore, after in vitro re-stimulation with MOG$_{35-55}$ for 48 h, splenocytes and splenic B cells from *Mb1cre Hif1af/f* mice produce a reduced level of IL-10 when compared to WT cells (Fig. 7d, e). However, there is no difference in TGF-β and IL-35 production by splenic B cells from *Mb1cre Hif1af/f* mice after re-stimulated by MOG$_{35-55}$ (Fig. 7e). Next, Th1 (IFN-γ⁺CD4⁺) cells, Th17 (IL-17⁺CD4⁺, IL-23R⁺IL-17⁺, or GM-CSF⁺IL-17⁺) cells, Treg (CD25⁺Foxp3⁺CD4⁺) cells, type 1 regulatory T cells (Tr1) (IL-10⁺CD4⁺) cells, IL-10⁺Foxp3⁺ T cells as well as IL-10⁺ B cells, ICAM⁺ B cells, CD73⁺ B cells, GITRL⁺ B cells, FasL⁺ B cells, and

PD-L1[+] B cells were quantified in the spleen, dLNs, and CNS by FACS. No difference is detected in the ICAM[+], CD73[+], GITRL[+], FasL[+], and PD-L1[+] B cell populations in spleen and dLNs (Supplementary Fig. 7b, c). Th1 and Th17 populations, including pathogenic IL-23R[+]IL-17[+] and GM-CSF[+]IL-17[+] Th17 cells are increased, whereas Tr1, IL-10[+]Foxp3[+] Treg cells, and IL-10-producing B cells are decreased in *Mb1^cre^Hif1a^f/f^* mice compared to WT mice (Fig. 7f–i and Supplementary Fig. 7d), suggesting that HIF-1α expression in B cells limits the progression of EAE by regulating Th1, Th17, and Tr1 cells differentiation. Collectively,

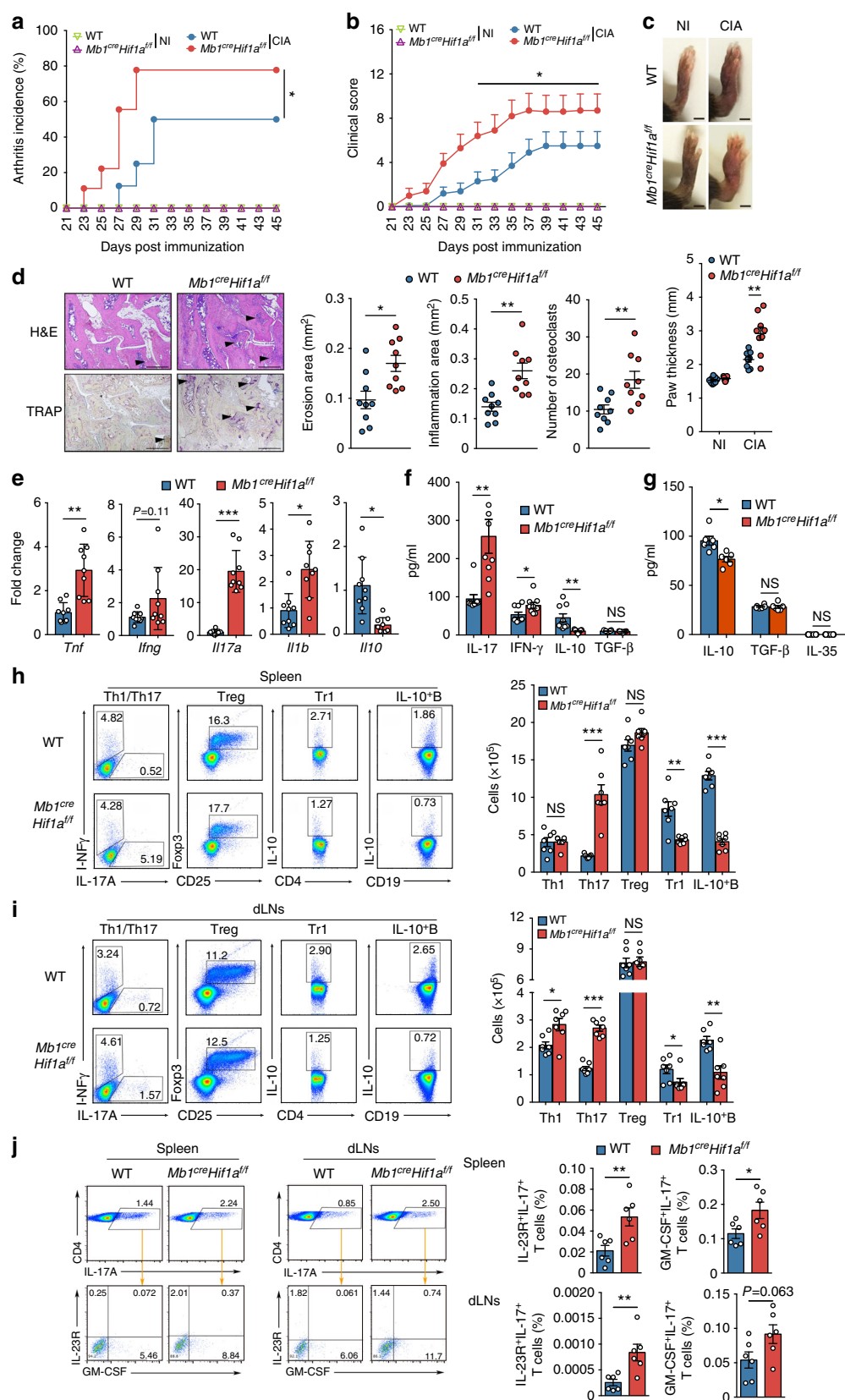

our results demonstrate that HIF-1α expression in B cells has a crucial protective function in autoimmune diseases.

**Impaired suppressive function of *Hif1a*-deficient B cells**. To determine whether the phenotypes of $Mb1^{cre}Hif1a^{f/f}$ mice in autoimmune diseases, including the observed exacerbated pro-inflammatory T cell response, were secondary to a defect in IL-10 production by B cells, naïve CD4 T cells were co-cultured with CD1d$^{hi}$CD5$^+$ B cells from WT mice in presence or absence of anti-IL-10 antibody. Indeed, CD4 T cells co-cultured with CD1d$^{hi}$CD5$^+$ B cells in presence of anti-IL-10 antibody shows higher Th1, Th17 and lower Tr1 polarization than the ones cultured in the absence of anti-IL-10 antibody (Fig. 8a). Next, naïve CD4 T cells were co-cultured with CD1d$^{hi}$CD5$^+$ B cells sorted from $Mb1^{cre}Hif1a^{f/f}$ mice and WT littermates under T cell-polarizing conditions. Naïve CD4 T cells co-cultured with CD1d$^{hi}$CD5$^+$ B cells from $Mb1^{cre}Hif1a^{f/f}$ mice (CD1d$^{hi}$CD5$^+$(ΔHif1a)) show higher polarization into Th1 (IFN-γ$^+$CD4$^+$) and Th17 (IL-17$^+$CD4$^+$) cells than their counter-parts co-cultured with sorted CD1d$^{hi}$CD5$^+$ B cells from WT mice (CD1d$^{hi}$CD5$^+$(WT)) (Fig. 8b). These data suggest that the suppressive function of *Hif1a*-deficient CD1d$^{hi}$CD5$^+$ B cells on Th1 and Th17 cells differentiation is impaired (Fig. 8b). Interestingly, less Tr1 (IL-10$^+$CD4$^+$) cells are also found when CD4 T cells where co-cultured with *Hif1a*-deficient CD1d$^{hi}$CD5$^+$ B cells, whereas no difference is detected in Foxp3$^+$ Treg cells (Fig. 8b, c), suggesting that CD1d$^{hi}$CD5$^+$ B cells regulate the differentiation of Tr1 cells in a HIF-1α-dependent manner in this culture system, as observed in vivo (Figs. 6 and 7).

This culture system was then used to define whether the impaired regulatory function of *Hif1a*-deficient CD1d$^{hi}$CD5$^+$ B cells was due to their defect in IL-10 production. To this end, *Hif1a*-deficient CD1d$^{hi}$CD5$^+$ B cells were transduced with an IL-10-overexpression lentivirus (Supplementary Fig. 8a), before co-culturing them with naïve CD4 T cells. Remarkably, the regulatory effects on Th1, Th17, and Tr1 cells differentiation are rescued for IL-10-transduced *Hif1a*-deficient CD1d$^{hi}$CD5$^+$ B cells (IL-10-CD1d$^{hi}$CD5$^+$(ΔHif1a)) compared to the mock-transduced *Hif1a*-deficient CD1d$^{hi}$CD5$^+$ B cells (mock-CD1d$^{hi}$CD5$^+$(ΔHif1a)) (Fig. 8b). These data indicate that HIF-1α-dependent IL-10 production is required for the suppressive function of CD1d$^{hi}$CD5$^+$ B cells on T cells in vitro.

We next used a similar genetic approach to test whether the suppressive defect of *Hif1a*-deficient CD1d$^{hi}$CD5$^+$ B cells in autoimmune disease could similarly be rescued by genetically ectopic-expressing IL-10. Thus, WT and *Hif1a*-deficient CD1d$^{hi}$CD5$^+$ B cells were sorted, transduced with mock or IL-10-overexpression lentivirus, and transferred into $Mb1^{cre}Hif1a^{f/f}$ mice that were subsequently immunized with MOG$_{35-55}$ to induce EAE (Supplementary Fig. 8b). Clinical score and pathological analyses of the spinal cord show that $Mb1^{cre}Hif1a^{f/f}$ mice

treated with CD1d$^{hi}$CD5$^+$(WT) cells have a significantly less severe disease than those treated with CD1d$^{hi}$CD5$^+$(ΔHif1a) cells (*$P < 0.05$ and **$P < 0.01$, by two-way analysis of variance with Bonferroni's post test and *t*-test; Fig. 9a, b). Remarkably, *Hif1a*-deficient CD1d$^{hi}$CD5$^+$ B cells transduced with the IL-10-overexpression lentivirus (IL-10-CD1d$^{hi}$CD5$^+$(ΔHif1a)) ameliorate the disease progression of $Mb1^{cre}Hif1a^{f/f}$ recipient mice as efficiently as CD1d$^{hi}$CD5$^+$(WT) cells (Fig. 9a, b). Similar results are obtained when analyzing the magnitude of the pro-inflammatory T cell response in these groups of mice. After in vitro re-stimulation with MOG$_{35-55}$ splenocytes isolated from mice administered with CD1d$^{hi}$CD5$^+$(WT) or IL-10-CD1d$^{hi}$CD5$^+$(ΔHif1a) cells show a lower level of IL-17, IFN-γ and a higher IL-10 level than the one's receiving CD1d$^{hi}$CD5$^+$(ΔHif1a) or mock-CD1d$^{hi}$CD5$^+$(ΔHif1a) cells (Fig. 9c). Accordingly, mice receiving CD1d$^{hi}$CD5$^+$(WT) or IL-10-CD1d$^{hi}$CD5$^+$(ΔHif1a) cells have reduced percentage and absolute cell number of Th1 or Th17 populations in dLNs than the mice receiving CD1d$^{hi}$CD5$^+$(ΔHif1a) or mock-CD1d$^{hi}$CD5$^+$(ΔHif1a) cells (Fig. 9d). Conversely, increased relative and absolute numbers of Tr1 cells are observed in dLNs from CD1d$^{hi}$CD5$^+$(WT) or IL-10-CD1d$^{hi}$CD5$^+$(ΔHif1a) recipient mice compared to CD1d$^{hi}$CD5$^+$(ΔHif1a) or mock-CD1d$^{hi}$CD5$^+$(ΔHif1a) recipients (Fig. 9d). Taken together, these findings show that the loss of HIF-1α in B cells causes impaired IL-10 production and aggravating autoimmune diseases (Supplementary Fig. 9).

## Discussion
Herein, we describe a novel molecular mechanism of immune modulation, which determines the function of IL-10-producing B cells and thereby influences the course of autoimmune disease. We identified HIF-1α as a critical transcriptional factor involved in IL-10 production by B cells, thereby influencing the course of T cell-mediated autoimmune diseases such as EAE and arthritis.

IL-10-producing B cells have been identified as potent players in the inhibition of inflammation in autoimmune disease[6]. Hence, it has been shown that adoptive transfer of B cells taken from DBA/1 mice in the remission phase of arthritis prevents the onset of CIA via the secretion of IL-10[7]. In accordance, transfer of B-cell activating factor of TNF family (BAFF) expanded CD1d$^{hi}$CD5$^+$ B cells decreased Th17 activation and reduced disease severity of arthritis[29]. In agreement with these data from arthritis models, adoptive transfer of MOG-sensitized CD1d$^{hi}$CD5$^+$ B cells into WT mice also mitigate the severity of EAE[8]. The immune regulatory function of CD1d$^{hi}$CD5$^+$ B cells appears to be tightly bound to the production of anti-inflammatory cytokines like IL-10, which limits the immune response to pathogens and thereby prevents damage to the host[30]. Studies have shown that CD1d$^{hi}$CD5$^+$ B cells have the property to differentiate into plasmablasts after stimulation[31]. Accordingly, we found that CD44$^{hi}$CD138$^+$ plasmablasts are also reduced in *Hif1a*-deficient

**Fig. 6** HIF-1α deficiency in B cells exacerbates collagen-induced arthritis. **a** Arthritis incidence in $Mb1^{cre}Hif1a^{f/f}$ ($n = 9$) and WT mice ($n = 9$) after collagen immunization. NI non-immunized mice; CIA collagen immunized mice. **b** Clinical score of arthritis in mice as described in **a**. **c** Picture and quantification of paw thickness at day 45 after the first immunization in mice as described in **a**. Scale bar, 2 mm. **d** Histopathology sections and quantifications of erosion area (H&E), inflammation area (H&E), and osteoclast number (TRAP) in paw from mice as in **a**. Arrows indicate erosion or inflammation area. Scale bars, 500 μm. **e** Quantitative RT-PCR analysis of *Tnf*, *Ifng*, *Il17a*, *Il1b*, and *Il10* mRNA expression in knee synovial tissue from mice as described in **a**. **f** IL-17, IFN-γ, IL-10, and TGF-β expression by splenocytes isolated from $Mb1^{cre}Hif1a^{f/f}$ ($n = 9$) and WT mice ($n = 9$) after collagen immunization followed by in vitro re-stimulation with collagen (20 μg/ml) for 48 h. **g** IL-10, TGF-β, and IL-35 production by enriched splenic B cells isolated from $Mb1^{cre}Hif1a^{f/f}$ ($n = 6$) and WT mice ($n = 6$) after collagen immunization followed by in vitro re-stimulation with collagen (20 μg/ml) for 48 h. **h, i** Representative plots and quantification of IL-17A$^+$CD4$^+$(Th17), IFN-γ$^+$CD4$^+$(Th1), CD25$^+$Foxp3$^+$CD4$^+$(Treg), IL-10$^+$CD4$^+$(Tr1), and IL-10$^+$CD19$^+$(IL-10$^+$B) cells in spleen (**h**) and draining lymph nodes (dLNs) (**i**) from $Mb1^{cre}Hif1a^{f/f}$ ($n = 7$) and WT mice ($n = 7$) after collagen immunization. **j** Representative plots and percentage of IL-23R$^+$IL-17A$^+$CD4$^+$ and GM-CSF$^+$IL-17A$^+$CD4$^+$T cells in spleen and dLNs from $Mb1^{cre}Hif1a^{f/f}$ ($n = 6$) and WT mice ($n = 6$) after collagen immunization. Data are shown as mean ± s.e.m. Pictures are representative of three independent experiments. NS not significant; *$P < 0.05$, **$P < 0.01$, and ***$P < 0.001$ (Kaplan–Meier analysis with log–rank test (**a**) or two-tailed unpaired Student's *t*-test (**b–h**)) (see also Supplementary Figure 6)

mice after EAE induction, suggesting that loss of HIF-1α causes impaired CD1d[hi]CD5[+] B cells and increases likelihood to develop autoimmune disease.

Published studies have shown that calcium sensor stromal interaction molecules (STIM) and IL-21-dependent cognate interactions are required for the function of IL-10-producing B cells[32,33]. However, the molecular regulation of IL-10 production in B cells was incompletely defined to date. Our data now show that HIF-1α is crucial in inducing IL-10 production by B cells. Lack of HIF-1α in B cells causes reduced IL-10 production followed by enhanced Th17 cells. In addition, increased IL-17 and IFN-γ production in *Hif1a*-deficient mice is associated with a

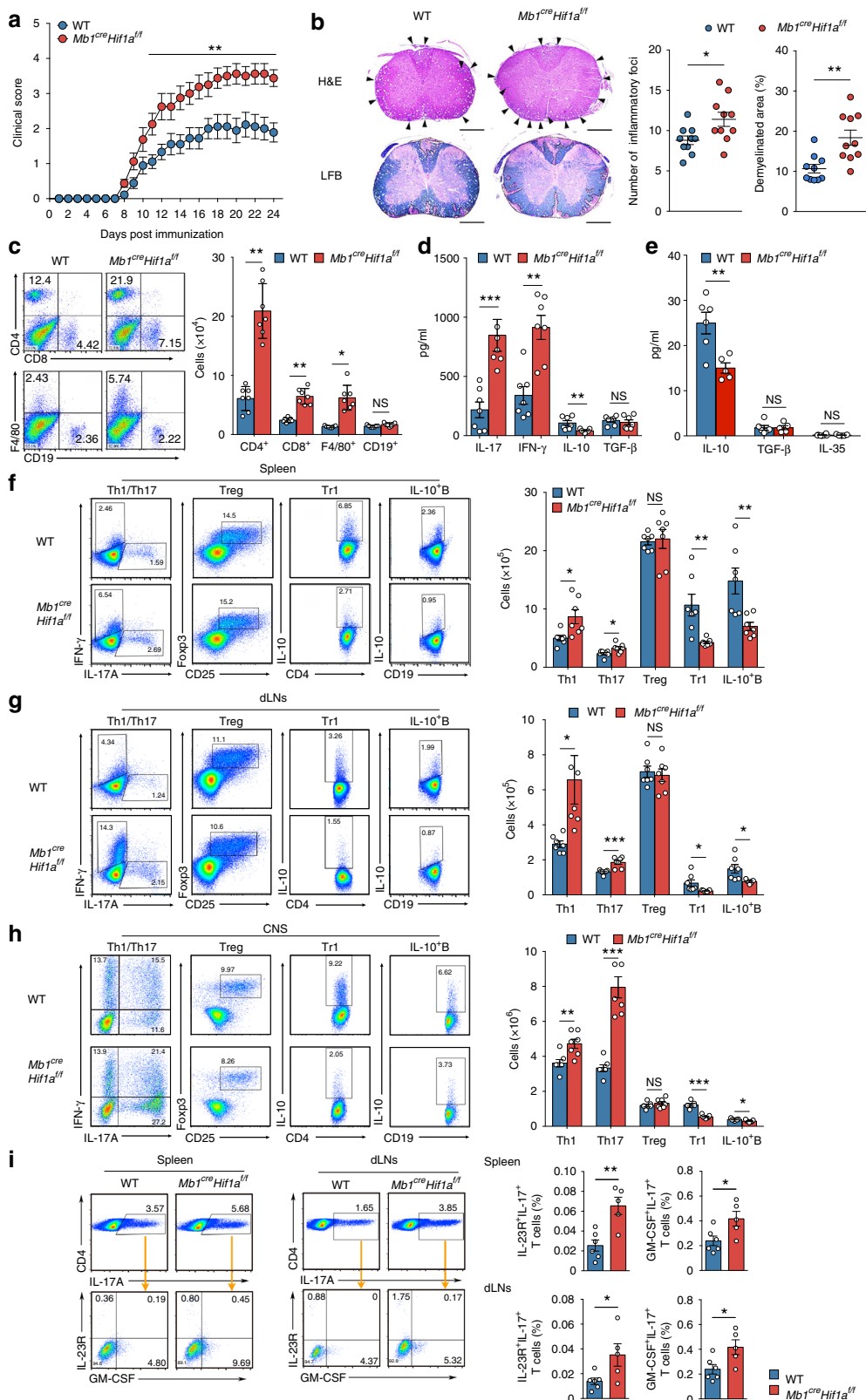

strong exacerbation of EAE and inflammatory arthritis. HIFs have been previously suggested to influence adaptive and innate immunity[34]. Differential effects of HIF-1α and HIF-2α have previously been shown in immune cells[14,16,17]. However, the roles of HIF-1α and HIF-2α in B cells have not been shown. Our data suggest that despite the description of hypoxic niches in the bone marrow and regions within the spleen[35], HIF-1α and HIF-2α appear non-essential for the development of B cell subsets in the bone marrow as well as spleens and lymph nodes in homeostasis. However, LPS or BCR-mediated activation of B cells causes massive induction of HIF-1α, but not HIF-2α, in an oxygen-independent way. While in macrophages, HIF-1α accumulation requires NF-κB dependent transcriptional event[19], HIF-1α in BCR-stimulated B cells is induced via the ERK-STAT3 signaling pathway. After B cell activation, STAT3 is phosphorylated at position Ser727, but not at the Thr705 site[36]. Our data show that phosphorylated-STAT3[727] then effectively induces Hif1a gene transcription in activated B cells.

Our findings indicate that HIF-1α contributes to CD1d$^{hi}$CD5$^+$ B cell proliferation and IL-10 production. Emerging studies indicate that metabolism is important for B cell function and proliferation[25]. In accordance with other publications[29,37,38], we found that CD1d$^{hi}$CD5$^+$ B cells have a higher glycolytic activity compared to CD1d$^{lo}$CD5$^-$B cells, which can control the normal expansion of CD1d$^{hi}$CD5$^+$ B cells. Our study shows for the first time that this glycolytic metabolism is dependent on HIF-1α expression. In addition, HIF-1α effectively binds to the Il10 promoter at two HRE elements, which also correlates with the actively transcribing regions. Since previous studies have shown that proximal broad H3K4me3 domains are highly dynamic in different cell types and conditions[39–41], the high enrichment of H3K4me3 in HRE I or HRE II regions is probably related to the hypoxic condition. We show that HIF-1α transcriptionally enhances Il10 mRNA expression. In line with our results, it has been shown that both IL-21 and IL-27 induce ERK and STAT3 activation and upregulate IL-10 in CD4 T cells[42,43]. Notably, HIF-1α appears to specifically induce IL-10 in B cells, as other immune regulatory factors expressed in B cells, such as TGF-β and IL-35, are not altered in Hif1a-deficient B cells. In accordance with our findings, the HIF-1α-dependent regulation of IL-10 has been previously reported in macrophages, Tr1 cells, and myocytes using different approaches of HIF-1α knockdown[44–46]. Our data also show that manipulation of HIF-1α expression in B cells influences Tr1 cells. Tr1 cells have been intimately associated with IL-10-producing B cells as they drive the differentiation of cognate B cells into IL-10-producing B cells[47], suggesting a mutual regulation between IL-10-producing B cells and Tr1 cells. In accordance with that, our data show that impaired IL-10 production by HIF-1α-dependent B cells is associated with decreased Tr1 cells, indicating a regulatory network between IL-10-producing B cells and Tr1 cells.

Therapeutically, fostering of HIF-1α expression may provide a tool to increase IL-10-producing B cells and limit autoimmune diseases such as EAE and arthritis. Inhibitors of prolyl-hydroxylases (PHD), for instance, are agents that can induce HIF-1α. Treatment with PHD inhibitors has been shown to ameliorate endotoxic shock as well as inflammatory bowel disease in mice[48–50]. In these studies, PHD inhibitors enhanced the numbers of IL-10-producing B cells and reduced expression of inflammatory cytokines[49]. Therefore, activating the HIF-1α axis through pharmacologic agents may indeed provide a tool to augment the immune regulatory potential of IL-10-producing B cells with the potential to prevent and/or treat systemic autoimmune inflammatory diseases[51].

In summary, we provide a novel molecular mechanism for the regulation of autoimmune disease by CD1d$^{hi}$CD5$^+$ B cells. By modulating glycolytic metabolism, HIF-1α regulates CD1d$^{hi}$CD5$^+$ B cell expansion. Moreover, we identified HIF-1α as a critical node involved in IL-10 production by B cells. HIF-1α effectively binds to hypoxia response elements on the Il10 promoter, resulting in expression of IL-10 in B cells. In consequence, HIF-1α expression in B cells regulates autoimmune diseases such as EAE and arthritis.

## Methods

**Mice.** C57BL/6 WT mice (027) were purchased from Charles River Laboratories (Sulzfeld, Germany). To generate B cell-specific Hif1a or Hif2a-deficient mice, Hif1a$^{flox/flox}$ mice or Hif2a$^{flox/flox}$ mice were crossed with Mb1-cre mice. Hif1a$^{flox/flox}$ mice, Hif2a$^{flox/flox}$ mice, and Mb1-cre mice were previously described[52–54]. The mice were bred and maintained on a C57BL/6 background and Hifs$^{flox/flox}$ cre-negative or Hifs$^{+/+}$ cre-positive littermates were used as WT controls. Sex- and age-matched (8–10 weeks) mice were killed using $CO_2$ and terminated via cervical dislocation for in vitro and ex vivo experiments. Animals were kept in a specific pathogen-free facility and animal experiments were approved by local ethics committee of Regierung von Mittelfranken (license Az: 55.2 2532-2-198 and 55.2 DMS-2532-2-84), Germany.

**Flow cytometry and cell sorting.** Single-cell suspensions were prepared from bone marrow (femurs), spleen, inguinal lymph nodes, peritoneal cavity, and peripheral blood. Red cells were lysed with ammonium-chloride-potassium (ACK) buffer. Cells were Fc-blocked (CD16/CD32) and stained with antibodies (Supplementary Table 1). Analyses of the expression of cell surface molecules on a single cell level were performed by flow cytometry with Calibur (BD) or Cytoflex (Beckman Coulter) flow cytometer. Dead cells were detected using a LIVE/DEAD Fixable Violet Dead Cell Stain Kit (L34955, Life Technologies) before cell surface staining. For analysis of intracellular IL-10 expression by B cells, LPS, phorbol-12-myristate-13-acetate (PMA), ionomycin, and monensin (L+PIM) were added to the cultures 5 h before fixing and permeabilizing with the Foxp3/Transcription Factor Staining Buffer Set (00/5523/00, ebioscience) according to the manufacturer's instruction. All flow cytometry experiments were gated on viable, single lymphocytes and data were analyzed with FlowJo software (Treestar).

For cell sorting, CD1d$^{lo}$CD5$^-$CD19$^+$B220$^+$ and CD1d$^{hi}$CD5$^+$CD19$^+$B220$^+$ B cells were sorted from splenocytes using a MoFlo cell sorter (DAKO instrumentations). Cell purity of 98–99% was generally achieved.

Gating strategies are presented in Supplementary Fig. 10.

---

**Fig. 7** HIF-1α deficient mice show exacerbated experimental autoimmune encephalomyelitis. **a** Clinical score of Mb1$^{cre}$Hif1a$^{f/f}$ (n = 10) and WT mice (n = 10) immunized with MOG$_{35-55}$. **b** Histopathology sections and quantifications of inflammatory loci (arrows) and demyelinated area (dashed line) in the spinal cord of Mb1$^{cre}$Hif1a$^{f/f}$ (n = 10) and WT mice (n = 10) showing lymphocyte infiltration (H&E) and demyelination area (LFB). Scale bars, 500 μm. **c** Representative plots and quantification of CNS-infiltrating cells in Mb1$^{cre}$Hif1a$^{f/f}$ (n = 7) and WT mice (n = 7) 18 days after induction of EAE. **d** IL-17, IFN-γ, IL-10, and TGF-β expression by splenocytes isolated from mice as described in **c** followed by an in vitro re-stimulation with MOG$_{35-55}$ (10 μM) for 48 h. **e** IL-10, TGF-β, and IL-35 production by enriched splenic B cells isolated from Mb1$^{cre}$Hif1a$^{f/f}$ (n = 5) and WT mice (n = 6) immunized with MOG$_{35-55}$ followed by in vitro re-stimulation with MOG$_{35-55}$ (10 μM) for 48 h. **f–h** Representative plots and quantification of IL-17A$^+$CD4$^+$(Th17), IFN-γ$^+$CD4$^+$(Th1), CD25$^+$Foxp3$^+$CD4$^+$(Treg), IL-10$^+$CD4$^+$(Tr1), and IL-10$^+$CD19$^+$(IL-10$^+$B) cells in spleen (**f**), draining lymph nodes (dLNs) (**g**), and CNS (**h**) from Mb1$^{cre}$Hif1a$^{f/f}$ (n = 7) and WT mice (n = 7) 18 days after induction of EAE. **i** Representative plots and percentages of IL-23R$^+$IL-17A$^+$CD4$^+$ and GM-CSF$^+$IL-17A$^+$CD4$^+$T cells in spleen and dLNs from Mb1$^{cre}$Hif1a$^{f/f}$ (n = 6) and WT mice (n = 6) 18 days after induction of EAE. Data are shown as mean ± s.e.m. Pictures are representative of three independent experiments. NS not significant; *P < 0.05, **P < 0.01, and ***P < 0.001 (unpaired, two-tailed Student's t-test) (see also Supplementary Figure 1)

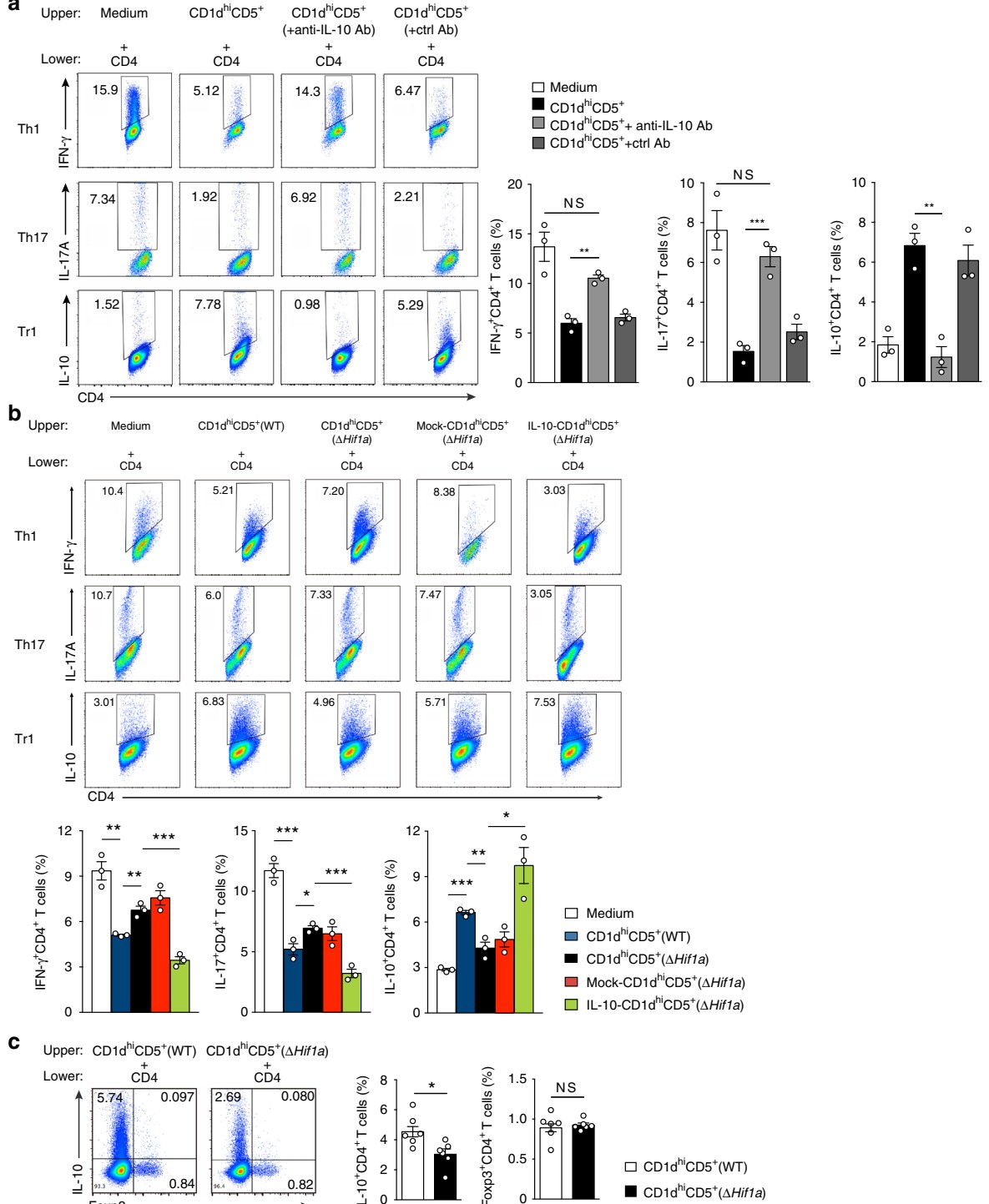

**Fig. 8** HIF-1α expression in B cells regulates helper T cell polarization in vitro. **a** Naïve CD4 T cells isolated from WT mice were co-cultured with CD1d^hiCD5^+ B cells from WT mice with anti-IL-10 antibody or isotype control antibody at a 1:1 T/B ratio in a trans-well system for 3 days under T cell-polarizing conditions. Percentage of IFN-γ^+CD4^+(Th1) cells, IL-17^+CD4^+(Th17) cells, and IL-10^+CD4^+ (Tr1) cells were determined (*n* = 3 per group). **b** Naïve CD4 T cells isolated from WT mice were co-cultured with CD1d^hiCD5^+ B cells from WT mice (CD1d^hiCD5^+(WT)), *Mb1^creHif1a^f/f* mice (CD1d^hiCD5^+(ΔHif1a)), *Mb1^creHif1a^f/f* mice after transduction of pDBR lentivirus (mock-CD1d^hiCD5^+(ΔHif1a)), or *Mb1^creHif1a^f/f* mice after transduction of pDBR-IL-10 lentivirus (IL-10-CD1d^hiCD5^+(ΔHif1a)) at a 1:1 T/B ratio in a trans-well system for 3 days under T cell-polarizing conditions. Percentage of IFN-γ^+CD4^+(Th1) cells, IL-17^+CD4^+(Th17) cells, and IL-10^+CD4^+(Tr1) cells were determined (*n* = 3 per group). **c** Naïve CD4 T cells isolated from WT mice were co-cultured with CD1d^hiCD5^+ B cells from WT mice (CD1d^hiCD5^+(WT)) or *Mb1^creHif1a^f/f* mice (CD1d^hiCD5^+(ΔHif1a)) at a 1:1 T/B ratio in a trans-well system for 3 days. Percentage of IL-10^+CD4^+ T cells and Foxp3^+CD4^+ T cells were determined (*n* = 6 per group). Data are shown as mean ± s.e.m. Pictures are representative of three independent experiments. NS not significant; *P < 0.05, **P < 0.01, and ***P < 0.001(unpaired, two-tailed Student's *t*-test)

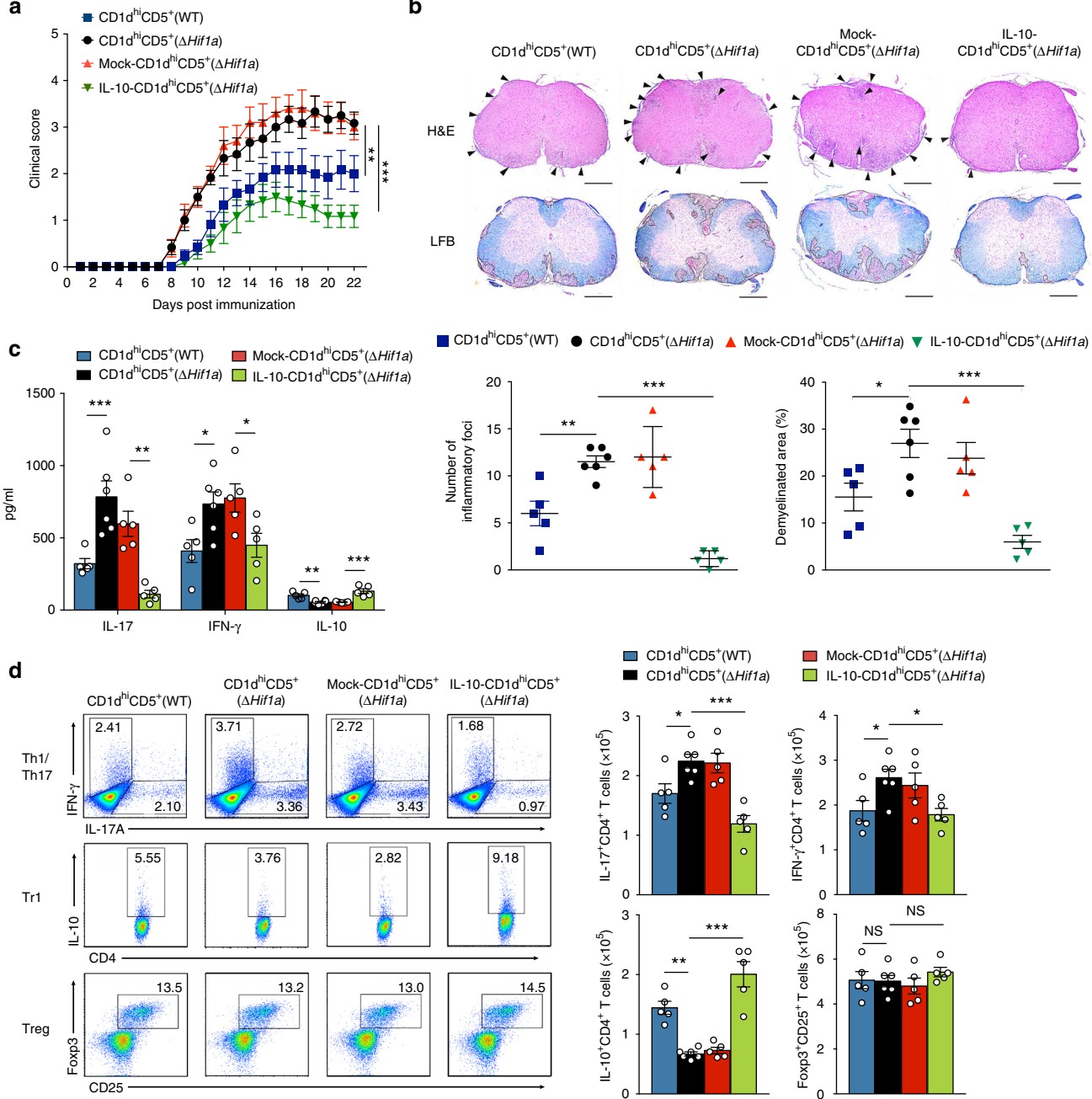

**Fig. 9** Aggravating EAE in *Hif1a*-deficient mice is rescued by ectopic expression of IL-10 in *Hif1a*-deficient CD1d^hiCD5^+ B cells. **a** Clinical score of *Mb1^creHif1a^f/f* mice immunized with MOG$_{35-55}$ after adoptive transfer of CD1d^hiCD5^+ B cells from WT mice (CD1d^hiCD5^+(WT)) (n = 5), *Mb1^creHif1a^f/f* mice (CD1d^hiCD5^+(ΔHif1a)) (n = 6), *Mb1^creHif1a^f/f* mice after transduction of pDBR lentivirus (mock-CD1d^hiCD5^+(ΔHif1a)) (n = 5), or *Mb1^creHif1a^f/f* mice after transduction of pDBR-IL-10 lentivirus (IL-10-CD1d^hiCD5^+(ΔHif1a)) (n = 5). **b** Pathophysiological pictures and quantifications of inflammatory loci (arrows) and demyelinated area (dashed line) in the spinal cord sections from *Mb1^creHif1a^f/f* mice immunized with MOG$_{35-55}$ after adoptive transfer of CD1d^hiCD5^+ (WT) (n = 5), CD1d^hiCD5^+(ΔHif1a) (n = 6), mock-CD1d^hiCD5^+(ΔHif1a) (n = 5), or IL-10-CD1d^hiCD5^+(ΔHif1a) cells (n = 5). Scale bars, 500 μm. **c** IL-17, IFN-γ, IL-10, and TGF-β expression by splenocytes isolated from *Mb1^creHif1a^f/f* mice immunized with MOG$_{35-55}$ after adoptive transfer of CD1d^hiCD5^+(WT) (n = 5), CD1d^hiCD5^+(ΔHif1a) (n = 6), mock-CD1d^hiCD5^+(ΔHif1a) (n = 5), or IL-10-CD1d^hiCD5^+(ΔHif1a) cells (n = 5) followed by in vitro re-stimulation with MOG$_{35-55}$ (10 μM) for 48 h. **d** Representative plots and quantification of IL-17^+CD4^+(Th17), IFN-γ^+CD4^+(Th1), IL-10^+CD4^+(Tr1), and CD25^+Foxp3^+CD4^+ (Treg) cells in draining lymph nodes from *Mb1^creHif1a^f/f* mice immunized with MOG$_{35-55}$ after adoptive transfer of CD1d^hiCD5^+(WT) (n = 5), CD1d^hiCD5^+(ΔHif1a) (n = 6), mock-CD1d^hiCD5^+(ΔHif1a) (n = 5), or IL-10-CD1d^hiCD5^+(ΔHif1a) cells (n = 5). Data are shown as mean ± s.e.m. Pictures are representative of three independent experiments. NS not significant; *P < 0.05, **P < 0.01, and ***P < 0.001 (two-way analysis of variance with Bonferroni's post-test (**a**) or two-tailed unpaired Student's *t*-test (**b**–**d**)) (see also Supplementary Figure 8)

**B cells isolation and stimulation.** For B cells isolation, resting B cells were purified from spleen by negative selection with anti-CD43 magnetic beads or positive selection with anti-B220 magnetic beads (MiltenyiBiotech) following the manufacturer's instructions. The purified B cells population was >95% B220 positive cells. B cells were cultured in RPMI supplemented with 10% FCS, 2-ME, penicillin (100 U/ml), and streptomycin (100 μg/ml). Purified B cells ($2 \times 10^6$/ml) were cultured with LPS (10 μg/ml) or IgM-specific goat F(ab')2 antibody (10 μg/ml) in a 48-well flat-bottom plate.

**Quantitative PCR analysis.** Total cell or tissue RNA was extracted using Trizol reagent (Invitrogen) and complementary DNA was synthesized by using High-Capacity cDNA Reverse Transcription Kit (4368814, Thermo Scientific) according to the manufacturer's instructions. Quantitative PCRs (QPCRs) were performed using SYBR Green I-dTTP (Eurogentec). Specific primers used for QPCR are listed in Supplementary Table 2. The levels of Hif1a, Hif2a, Tgfb, Il10, P35, Ebi3, Tnf, Il17, Ifng, Il1b, Glut, PkM2, Hk2, Ldha, Pdk1, and Gpi1 were determined by evaluating the threshold cycle (Ct) of target gene after normalization against the Ct value of Hprt and calculated using the formula $2^{-(\text{Ct of target gene} - \text{Ct of } Hprt)}$.

**Western blot and co-immunoprecipitation analysis.** Cultured B cells were washed twice with PBS and homogenized into extraction buffer (8 M urea, 10% glycerol, 1% SDS, 10 mM Tris-HCl pH 6.8, protease inhibitor complete (Roche), 1 mM Sodium-Vanadate). Total cell lysates were resolved on 10% SDS–PAGE and were transferred to nitrocellulose membrane (Bio-Rad). The following primary antibodies were used (Supplementary Table 1): HIF-1α antibody, HIF-2α antibody (Novus), STAT3 phosphorylated at Ser727 and Thr705 antibodies, total STAT3 antibody, phosphorylated ERK antibody, total ERK antibody (Cell Signaling), and β-actin antibody (Sigma). The western blot bands were quantified using ImageJ Software.

The Dynabeads co-immunoprecipitation kit (14321D, Invitrogen) was used for the endogenous co-immunoprecipitation assay and nuclear extracts preparation was described previously[55]. Briefly, splenic B cells were enriched from WT mice and stimulated with anti-IgM (10 μg/ml) for 8 h. Five milligrams of nuclear extracts were incubated with 5 μg of anti-HIF-1α antibody (H1α67), anti-pSTAT3[727] antibody, or control IgG with Dynabeads protein G according to the manufacturer's instruction. Immunoblots were then performed using anti-HIF-1α or anti-STAT3 phosphorylated at Ser727 as described above.

Full size images are presented in Supplementary Fig. 11.

**Luciferase reporter assay.** HRE regions (I–V) of Il10 promoter (Supplementary table 2) were amplified by PCR from genomic DNA extracted from splenocytes in C57BL/6 WT mice and cloned into the pGL3 firefly reporter vector (Promega). 293T cells were co-transfected with luciferase reporter construct and β-gal plasmid using Lipofectamine 2000 (invitrogen). Transfected cells were cultured under normoxic (21% $O_2$) or hypoxic (1% $O_2$) conditions for 24 h. Cells were then lysed and luciferase activity was quantified and normalized to the activity of the co-transfected β-gal reporter gene.

**RNA interference.** STAT3, ERK, and RelA, or scrambled siRNA lentivirus constructs were obtained from Applied Biological Materials (ABM, Richmond, BC, Canada). Transduced splenic B cells were incubated with 10 μg/ml anti-IgM or LPS for 4 h and then cell lysates was used for western blot.

**Chromatin immunoprecipitation.** Splenic B cells were enriched from C57BL/6 WT mice and cultured in medium alone or anti-IgM (10 μg/ml) for 8 h. Next, ChIP experiments were performed with ChIP-IT Express kit (53018, Active Motif) according to the manufacturer's protocol. Ten micrograms of anti-pSTAT3[727], anti-HIF-1α, anti-HIF-1β, or anti-HIF-2α antibodies, as well as control IgG antibody, were used for the immunoprecipitation. Primer sequence for STAT3 binding site was described previously[16] (Supplementary Table 2). HIF-1α binding sites were predicted by JASPAR with the consensus core (A/GCGTG) and primers were designed by Primer-BLAST (Supplementary Table 2).

**Glucose uptake and lactate production assays.** Glucose uptake was determined using Glucose uptake cell-based assay kit (600470, Cayman) according to the manufacturer's protocol. Lactate production was determined in the supernatant collected from sorted CD1d$^{hi}$CD5$^+$CD19$^+$ B cells with or without stimulation for 6 h using the Lactate colorimetric/fluorometric assay kit (K607, Biovision) according to the manufacturer's protocol.

**T cells co-cultures and polarization assay.** Naïve CD4 T cells were enriched from the spleen of WT mice using MACS beads (130-104-453, Miltenyi Biotech) and cultured with plated-bound anti-CD3 (5 μg/ml) and soluble anti-CD28 (2 μg/ml) (eBioscience). Cytokines and neutralizing antibodies were added at the following concentrations: (Th1) 5 ng/ml IL-2, 10 ng/ml IL-12 (Peprotech), 10 μg/ml anti-IL-4 (ebiosciences); (Th17) 5 ng/ml IL-2, 5 ng/ml TGF-β, 10 ng/ml IL-6 (Peprotech); for Tr1 conditions, only 5 ng/ml IL-2 was added. Enriched CD4 T cells ($1 \times 10^5$) were plated in the bottom compartment of the cell culture wells

while sorted CD1d$^{hi}$CD5$^+$ B cells ($1 \times 10^5$) were cultured in the trans-well inserts (0.4 μm pore size; Corning). After 3 days, CD4$^+$ T cells were stimulated for 5 hours with PMA and ionomycin in the presence of monensin for intracellular cytokine staining.

**In vitro suppression assay.** Splenic CD4$^+$CD25$^-$ Teff and CD4$^+$CD25$^+$ Treg cells were sorted by MoFlo cell sorter (DAKO instrumentations). Purities of 98–99% were generally achieved. CD4$^+$CD25$^-$ Teff cells were labeled with 5 μM CFSE for 10 min at 37 °C, co-cultured with sorted CD4$^+$CD25$^+$ Treg cells for 72 h in flat bottom 96-well plates.

**Proliferation and apoptosis assay.** Splenic B cells were enriched by CD43 magnetic beads (Militenyi Biotec). The purified B cell population was >98% positive for B220 staining. For proliferation assays, enriched or sorted B cells were labeled with 5 μM CFSE or Celltrace violet (Thermo Scientific) at 37 °C for 10 min and stimulated with anti-IgM, anti-CD40, or LPS for 72 h. For apoptosis assay, enriched B cells were stimulated with anti-IgM, anti-CD40, or LPS for 48 h and then stained with TOPRO (Thermo Scientific) and Annexin-V (ebioscience). For toxic assays of inhibitors, enriched B cells were stimulated with anti-IgM for 4 h with or without doses of inhibitors and then stained with TOPRO and Annexin-V.

**Immunization and enzyme-linked immunosorbent assay.** Mice were injected intraperitoneally (i.p.) with 100 μg of NP-CGG in alum, 50 μg of NP-LPS (Biosearch Technologies), or 50 μg NP-Ficoll (Biosearch Technologies). For secondary immunization, mice were injected i.p. with 50 μg NP-CGG without adjuvant. Levels of NP-specific antibodies were analyzed on 96-well plates coated with 5 μg/ml NP(20)-BSA (Biosearch Technology) and detected with HRP-conjugated goat anti-mouse IgG3, IgG1, and IgM antibodies (SouthernBiotech). For measurement of cytokine release of auto antigen-reactive lymphocytes, single-cell suspensions of splenocytes or enriched splenic B cells prepared from mice after CIA or EAE induction were cultured with CII (20 μg/ml) or MOG$_{35-55}$ (10 μM) for 48 h. IL-17, IFN-γ, IL-10, and TGF-β levels in the culture medium were detected by ELISA according to the manufacturer's protocol (DY421, DY485, DY417, and DY1679, R&D). Alternatively, IL-17, IFN-γ, IL-10, IL-6, IL-35, and TGF-β levels in the serum from mice 18 days after EAE induction were detected by ELISA according to the manufacturer's protocol (DY421, DY485, DY417, DY1679, and DY406, R&D; 440507, Biolegend).

**In vivo proliferation assay.** Mice were fed with 0.8 mg/ml BrdU in drinking water during 7 days before analysis. Cells were then stained with indicated antibodies and prepared according to the BrdU Flow kit (559619, BD Pharmingen) protocol.

**CIA and EAE animal models.** CIA: Mb1$^{cre}$Hif1a$^{f/f}$ mice and WT littermates (8–12 weeks of age) were immunized by intradermal injection in the tail with 100 μg of chicken type II collagen (CII) in complete Freunds' adjuvant (CFA) (Sigma). Twenty-one days after the primary immunization, the mice were boosted with a secondary immunization with same amount of CII emulsified in incomplete Freunds' adjuvant (Sigma) intradermally in the tail proximal to the primary injection site. Next, the clinical scores for each paw were evaluated every other day and scored individually on a scale of 0–4, which results in a maximum score of 16. Each paw is scored as follows: 0, no evidence of erythema and swelling; 1, erythema and mild swelling confined to the tarsals or ankle joint; 2, erythema and mild swelling extending from the ankle to the tarsals; 3, erythema and moderate swelling extending from the ankle to metatarsal joints; 4, erythema and severe swelling encompass the ankle, foot, and digits, or ankylosis of the limb.

EAE: Mb1$^{cre}$Hif1a$^{f/f}$ mice and WT littermates were immunized subcutaneously with 100 μg MOG peptide$_{35-55}$ (Charité Berlin) in 50 μl $H_2O$ emulsified in 50 μl CFA, which was enriched with 10 mg/ml Mycobacterium tuberculosis (H37Ra, Difco/PD PharMingen) on day 0 in order to induce EAE. In addition, 200 ng pertussis toxin (List/Quadratech) was administered i.p. on day 0 and day 2. EAE paralysis of mice was scored as follows: 0, no disease; 1, tail weakness; 2, paraparesis; 3, paraplegia; 4, paraplegia with forelimb weakness; 5, moribund or death.

**Histology.** On day 45 of CIA model, whole paw joints were fixed in 4% paraformaldehyde, decalcified in EDTA, and then embedded in paraffin. Specimens were longitudinally cut into 4 μm sections, then hematoxylin and eosin (H&E) and tartrate-resistant acid phosphatase (TRAP) stainings were performed.

After completing the EAE experiments, spinal cords were dissected from mice after transcardially perfused with 4% paraformaldehyde and post-fixed them overnight. Paraffin-embedded sections (8 μm) of spinal cords were then stained with H&E and luxol fast blue (LFB) staining.

**Isolation of CNS-infiltrated lymphocytes.** After cardiac perfusion with PBS, CNS tissues were digested with 2.5 mg/ml collagenase D (Roche) and 1 mg/ml DNaseI (Roche) at 37 °C for 45 min. Lymphocytes were isolated by passing the tissue through 70 μm cell strainers, followed by Percoll (Millipore) gradient (70%/37%) centrifugation. Lymphocytes were collected from the interface and washed in PBS.

**Lentivirus transfection and adoptive transfer**. Production of viral supernatants and B cell transduction were described previously[56]. Briefly, lentiviral particles were produced in 293T cells by co-transfection of psPax2 packaging vector (Addgene), VSVG envelope plasmid (Addgene), pDBR (mock), or pDBR-IL-10 plasmid using Lipofectamine 2000 (Invitrogen). After 48 h, supernatants were collected, filtered (0.45 μm), and supplemented with 10 mM HEPES (Invitrogen) and 10 μg/ml polybrene (Millipore). Sorted CD1d$^{hi}$CD5$^+$ B cells were centrifuged at $3.5 \times 10^6$/ well in six-well plates with 3 ml of viral supernatants in a total volume of 4 ml at 1800 rpm during 75 min at room temperature, and then washed in PBS. A total of $1.5 \times 10^6$ transduced CD1d$^{hi}$CD5$^+$ B cells were transferred intravenously into recipient mice 24 h before EAE induction.

**Statistical analysis**. For comparison of the two groups, linear regression with a 95% confidence interval, and two-tailed Student's $t$-test were used. Kaplan–Meier analysis with log-rank test was used to determine the significance of CIA incidence. Two-way analysis of variance with Bonferroni's post test for paired data was used to determine the significance of EAE clinical scores in adoptive transfer experiment. GraphPad Prism software 6.0 was used for statistical analysis. $P$-value of less than 0.05 was considered statistically significant.

**Data availability**. The authors declare that all data supporting the findings of this study are available within the article and its Supplementary Information files or are available from the authors on request.

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

## Acknowledgements

We are very grateful to Uwe Appelt for the help with cell sorting and to Dr. Dirk Mielenz and Dr. Tobias Rothe for technical advises. This study was supported by the Deutsche Forschungsgemeinschaft (CRC1181-A01, SPP1468, IMMUNOBONE; BO3811/1-1, Emmy Noether, BO 3811/6-1).

## Author contributions

X.M. and A.B. designed the study and wrote the manuscript; X.M., B.G., and K.X.K. carried out the in vitro experiments; X.M. and Y.L. carried out the in vivo experiments; M.S.W., X.-X.C., J.J., and S.F. contributed to discussion and manuscript preparation; G.S. and A.B. supervised the study and edited the manuscript.

## Additional information

**Competing interests:** The authors declare no competing financial interests.

