## [Peer Review File · Nature Communications]

Reviewers' comments (23.10.2017):

Reviewer #1 (Remarks to the Author):

The authors performed a comprehensive analysis of the immunological consequences of HIF-1alpha loss of function in B lymphocytes. They show that HIF-1-dependent expression of interleukin 10 is critical for the function of regulatory B cells and that conditional knockout of HIF-1alpha in B cells results in exacerbation of autoimmunity. The work is novel, interesting, and the data strongly support the authors' major conclusions. I have only two minor comments:

1. The authors should distinguish between a HIF-1 binding site, as identified by chromatin immunoprecipitation (ChIP) assays, and a hypoxia-response element as determined by luciferase reporter assays. With regards to the ChIP assays, it would be useful to show that binding of HIF-1 beta is specifically induced at the same sites as HIF-1alpha (since the heterodimer is the functional unit for DNA binding), whereas HIF-2alpha binding is not induced. With respect to the luciferase assay, the authors should state precisely what sequences were included in the reporter constructs, as the 5-bp consensus binding site is not sufficient for HRE activity.

2. The authors may want to note that HIF-1alpha-dependent regulation of IL-10 has previously been reported: K. Sarkar et al. Proc Natl Acad Sci USA. 2012;109(26):10504.

Reviewer #2: (Remarks to the Author):

In this manuscript, Meng and collaborators describe a potential role that HIF-1a has in the transcription of IL-10 in Bregs. The results are interesting and the work is well executed. However, there several major points that need to be addressed.

1) Fig 1 uses wild-type splenic B cells, either untreated or stimulated with LPS or IgM, in all experiments – not sure how relevant this is to regulatory B cells?

2) Fig 1a shows mRNA induction of HIF 1a, but not HIF2a, in response to LPS and IgM. It would be easier to interpret the fold change if the zero hour time point was used as a calibrator and therefore set to one. It is not clear why the authors chose a concentration of 10ug/ml of LPS and IgM to stimulate B cells?

3) Figure legend 1a states “Data represent mean±s.e.m. from two or more independent experiments (a-e)”. Could the authors be more specific? It would be useful to indicate how many independent experiments were performed per experiment for each experiment in each figure.

4) For Fig 1d, is there any cytotoxicity data to show the inhibitors of STAT3, ERK or AKT were not killing the cells? Also, it is not explained why these inhibitors were chosen, or why complementary/more specific methods such as siRNA or overexpression analyses were not used.

5) On page 6, the authors state “Since HIF-1 α induction by LPS has been already reported to be dependent on NF- κ B signaling¹⁸”. However, the reference they cite refers to a study in human monocytes. What evidence do the authors have that the signalling pathways utilised are the same in these different cells/species?

6) Fig 1e. How did the authors identify the putative STAT3 binding site in the Hif1a promoter? Also, it would be better to present the data as %input rather than fold enrichment.

The reduction in IL-10 production, observed in figure 4 in conditional KO mice is very modest. Perhaps the authors should try other way to stimulate B cells to asses whether they see a better reduction.

7) Supplementary figure 4. It would be easier to interpret the fold change if the wild-type sample was used as a calibrator and therefore set to one.

8) Figure 4d. It would be easier to interpret the fold change if the wild-type sample was used as a calibrator and therefore set to one.

9) Figure 5b. How were the putative HRE identified?

10) Figures 5c & d. 5c shows HIF-1a binding specifically to sites I and II in the IL10 promoter, which are located about 3kb upstream of the TSS. And the pattern of H3K4me3 was similar. I have difficulty understanding why this is the case, since H3K4me3 should broadly mark actively transcribing promoters, not just mirror the binding of specific transcription factors. Also, the data would be better presented as %input.

11) Figures 5g & h. The data would be better presented as %input.

12) The results in figure 6 are very convincing and it is clear that the lack of HIF1a in B cells has an effect on disease pathogenesis. However, this does not reflect the rather limited effect that it has on IL-10 production, suggesting that Bregs may suppress via other mechanisms. The authors should check at least all the IL-10 independent mediated mechanisms previously reported in the literature.

13) Figure 8 shows the suppressive capacity of Hif-1a-deficient B cells are rescued by overexpression of IL10. This system is very artificial and it is likely that forced overexpression of IL10 in any B cell subset would result in a suppressive phenotype.

Reviewer #3 (Remarks to the Author):

This is a well-conducted study that provides strong evidence, using a variety of strategies that HIF-1a contributes to, but is not essential, for IL-10 production by B cells. While all B cell subsets can produce IL-10, they don't all regulate the severity of autoimmune disease in mice. Thus the study focuses on HIF-1a in CD1dhiCD5+ B cells. However, IL-10 production in CD1dhiCD5+ HIF-1a-deficient B cells was not shown. A reduction in IL-10 production is critical to the interpretation of the data shown. The authors blur the lines between IL-10 production by B cells and regulation of inflammation by B cells producing IL-10. B cells do not switch into an IL-10-producing regulatory B cell subset. They are regulatory because they have the capacity to produce high levels of IL-10. The two are not synonymous. A connection between HIF-1a and IL-10 production in macrophages has already been reported. In addition, a role for HIF-1a in metabolism in macrophages and Tr1 cells has also been reported. These studies should have been discussed and thus reduce the novelty of the current study.

1. The article incorrectly assumes that B cells differentiate or switch into IL-10 producing cells (page 4, line 56). All B cell subsets can produce IL-10 given the appropriate stimuli. There is no "switching" into a regulatory pathway. Thus there is no "official" IL-10-producing regulatory B cell subset as inferred in this manuscript. The authors are encouraged to read the following review for clarification: PMID: 25663677.

2. For all figures, please provide the number of mice or data points represent in the line and bar graphs. The number of independent experiments is not sufficient.

3. How do the authors explain that there is a reduction in T2-MZP, but not a subsequent reduction in MZ B cells? T2-MZP are not regulatory B cells. They are the precursor to MZ B cells. Likewise, CD1dhiCD5+ B cells are a subset of PC B cells and not a separate lineage.

4. Thus there is no defect in regulatory B cell proliferation as stated on lines 154, 163. All the data show in Fig. 3d is that the two cell populations in the steady state have a reduction in the number of cells cycling. Hif1a is known to regulate cell proliferation, so its deletion is consistent with a reduction in cell proliferation, which has nothing to do with a regulatory B cell phenotype or their specific regulation by Hif1a. It would be interesting to know whether the results in Fig. 4 are specific to CD1dhiCD5+ B cells or whether similar results would be obtained in other B cells subsets. Without that information, nothing can be concluded about the specificity of Hif1a for CD1dhiCD5+ B cells.

5. It is unclear why some experiments are performed using total B cells and others are performed using specific B cell subsets. This makes the data difficult to interpret in regards to B cell regulation by HIF-1a.

6. In Fig. 8a, the phenotype of the CD4+IL-10+ T cells should be identified. Certain B cell subsets can support iTreg (CD4+Foxp3+) generation. This could explain much of the data in Fig. 8. IL-10+CD4+ is also a phenotype consistent with Foxp3+ T regulatory cells.

7. There is very little evidence that IL-10 directly suppresses CD4 T cell function as suggested in Fig. 8, line 276. To make that claim, IL-10 would have to be neutralized in the culture and IL-10R-deficient T cells would need to be used.

8. Fig. 8. No data is provided showing that the IL-10-CD1dhiCD5+ Hif1a-/- B cells actually produce higher levels of IL-10 as compared to controls.

9. Line 358, no definitive data was shown that HIF-1a-dependent B cells are crucial for the differentiation and function of Tr1 cells. Tr1 differentiation and function were not studied.

10. The authors conclude that HIF-1a is crucial in inducing IL-10 production in B cells. Since all B cell subsets produce IL-10 phrasing such as .."suggesting that HIF-1 is essential for all subsets of IL-10-producing B cells" is not quite accurate and blurs the actual results.

11. It was never demonstrated that IL-10 production in CD1dhiCD5+ Hif1a-/- B cells is reduced as compared to controls.

Minor comments

12. Line 54, 82 and 328, encephalomyelitis and experimental autoimmune encephalomyelitis should be "experimental autoimmune encephalomyelitis".

13. Journal names in the reference list were not correctly abbreviated.

14. The sentence ending in line 315 needs to be referenced.

15. Line 326, no evidence was provided that the IL-17 producing cells are pathogenic. Many of the experiments were performed using splenocytes, that contain a number of different cell types that can contribute to IL-10 production.

16. There are English syntax errors that need to be corrected.

Response to Reviewers (7.11.2017)

Re: NCOMMS-17-12195

The following corrections were made:

Reviewers' comments:

Reviewer #1

The authors performed a comprehensive analysis of the immunological consequences of HIF-1alpha loss of function in B lymphocytes. They show that HIF-1-dependent expression of interleukin 10 is critical for the function of regulatory B cells and that conditional knockout of HIF-1alpha in B cells results in exacerbation of autoimmunity. The work is novel, interesting, and the data strongly support the authors' major conclusions. I have only two minor comments:

1. The authors should distinguish between a HIF-1 binding site, as identified by chromatin immunoprecipitation (ChIP) assays, and a hypoxia-response element as determined by luciferase reporter assays. With regards to the ChIP assays, it would be useful to show that binding of HIF-1 beta is specifically induced at the same sites as HIF-1alpha (since the heterodimer is the functional unit for DNA binding), whereas HIF-2alpha binding is not induced. With respect to the luciferase assay, the authors should state precisely what sequences were included in the reporter constructs, as the 5-bp consensus binding site is not sufficient for HRE activity.

We thank the reviewer for these valuable comments. The HRE regions on IL-10 promoter were predicted by JASPAR with the consensus core A/GCGTG (Figure 5b and Supplementary Figure 5a). For luciferase reporter assay, we have sub-cloned 100-120bp genomic sequence around these HRE regions according to the method (Zhen Chen et,al J Clin Invest. 2013;123(3):1057-1067). We have detailed the sequences information in supplementary table 2. In addition, we have performed ChIP analysis for HIF-1 β and HIF-2 α in BCR-stimulated B cells. As shown in supplementary figure 5b and c, HIF-1 β could bind to the HRE I and HRE II regions of the *IL10* promoter whereas HIF-2 α could not bind the *IL10* promoter, indicating that HIF-1 heterodimers regulate *IL10* transcription in activated B cells.

2. The authors may want to note that HIF-1alpha-dependent regulation of IL-10 has previously been reported: K. Sarkar et al. Proc Natl Acad Sci USA. 2012;109(26):10504.

As suggested by the reviewer, we have included and discussed this reference (page19).

Reviewer #2:

In this manuscript, Meng and collaborators describe a potential role that HIF-1a has in the transcription of IL-10 in Bregs. The results are interesting and the work is well executed. However, there several major points that need to be addressed.

1) Fig 1 uses wild-type splenic B cells, either untreated or stimulated with LPS or IgM, in all experiments – not sure how relevant this is to regulatory B cells?

In B cells, several stimulations are required to study their state of development and activity. We have chosen to analyze B cells following LPS or α IgM stimulation, because a reduction in the number of IL-10 competent B cells is seen in mice when CD19, a strong positive regulator of BCR signaling, is knocked out. Furthermore, preliminary studies suggested that lipopolysaccharide (LPS) stimulation promotes the production of IL-10 by competent B cells (Tedder TF, J Immunol 2015; 194:1395-1401). Therefore stimulations, mainly in the form of BCR stimulation, are required for IL-10 by B cells.

2) Fig 1a shows mRNA induction of HIF 1a, but not HIF2a, in response to LPS and IgM. It would be easier to interpret the fold change if the zero hour time point was used as a calibrator and therefore set to one. It is not clear why the authors chose a concentration of 10ug/ml of LPS and IgM to stimulate B cells?

As suggested by the reviewer, we have interpreted the data in fold change with time zero set to 1. We have used the concentration of 10 mg/mL for B cells stimulation based on previous reports showing the efficacy of this dose (Yanaba K et al, J Immunol. 2009; 182: 7459–72; Horikawa M et al, J Clin Invest. 2011;121:4268–80).

3) Figure legend 1a states “Data represent mean \pm s.e.m. from two or more independent experiments (a-e)”. Could the authors be more specific? It would be useful to indicate how many independent experiments were performed per experiment for each experiment in each figure.

As requested by the reviewer, we have repeated some experiments and clearly state the number of independent experiments for all figures.

4) For Fig 1d, is there any cytotoxicity data to show the inhibitors of STAT3, ERK or AKT were not killing the cells? Also, it is not explained why these inhibitors were chosen, or why complementary/more specific methods such as siRNA or overexpression analyses were not used.

As suggested by the reviewer, Annexin-V and PI stainings were performed in activated splenic B cells following treatment with STAT3, ERK or AKT inhibitors. As shown in Supplementary figure 1b, no toxic effect could be detected following the inhibitor treatments. Moreover, we

have completed our analyses with siRNA approaches to knock-down STAT3 or ERK. As shown in supplementary figure 1c, HIF-1 α expression was decreased when STAT3 and ERK were knocked down by lentivirus transfection.

5) On page 6, the authors state “Since HIF-1 α induction by LPS has been already reported to be dependent on NF- κ B signaling¹⁸”. However, the reference they cite refers to a study in human monocytes. What evidence do the authors have that the signalling pathways utilised are the same in these different cells/species?

As requested by the reviewer, we have performed lentivirus transfection to knock-down RelA gene, an essential protein for NF- κ B signaling, in LPS stimulated B cells. As shown in supplementary figure 1d, HIF-1 α protein level was decreased when RelA was knocked down by siRNA, suggesting that the NF- κ B signaling is also important in the control of HIF-1 α protein levels.

6) Fig 1e. How did the authors identify the putative STAT3 binding site in the *Hif1a* promoter? Also, it would be better to present the data as %input rather than fold enrichment.

The reviewer was right. Indeed, we have used preliminary described putative STAT3 binding site in *Hif1a* promoter (Dang EV et al, Cell 2011;146:772-84). We have made this clear in the materials and methods part and the supplementary table 2. As suggested by the reviewer, we have presented the ChIP data as % of input.

The reduction in IL-10 production, observed in figure 4 in conditional KO mice is very modest. Perhaps the authors should try other way to stimulate B cells to asses whether they see a better reduction.

We used LPS+PIM (PMA, ionomycin, plus monensin) because they were previously used in many studies to stimulate IL-10 production by B cells (Maseda D et al, J Immunol. 2013; 191: 2780–95; Yoshizaki A et al, Nature. 2012 8;491:264-8). Moreover, after gating on CD1d^{hi}CD5⁺ population, the reduction for IL-10 production in *Hif1a*-deficient B cells was stronger than in total B cells (Figure 3d). In addition, we used hypoxia as another stimulation of the HIF-1 α -IL10 pathway. Following 24h of hypoxic culture, there was a highly significant impairment of IL-10 upregulation in *Hif1a*-deficient compared to WT B cells (Figure 5a).

7) Supplementary figure 4. It would be easier to interpret the fold change if the wild-type sample was used as a calibrator and therefore set to one.

See 8

8) Figure 4d. It would be easier to interpret the fold change if the wild-type sample was used as a calibrator and therefore set to one.

As suggested by the reviewer, mRNA quantifications were now presented as fold change.

9) Figure 5b. How were the putative HRE identified?

The putative HRE regions on *Il10* promoter were predicted by JASPAR with the consensus core A/GCGTG (Supplementary Figure 5a). We have stated this in the materials and methods part and supplementary table 2.

10) Figures 5c & d. 5c shows HIF-1a binding specifically to sites I and II in the IL10 promoter, which are located about 3kb upstream of the TSS. And the pattern of H3K4me3 was similar. I have difficulty understanding why this is the case, since H3K4me3 should broadly mark actively transcribing promoters, not just mirror the binding of specific transcription factors. Also, the data would be better presented as %input.

The reviewer is right, other regions with high level of H3K4me3 in IL-10 promoter may exist. However, we have focused on the regions, where HIF-1 α could bind IL10 promoter to determine whether these regions could also be transcriptionally active. We have discussed this point in page 19. As requested by the reviewer, we have changed the ChIP data in % of input.

11) Figures 5g & h. The data would be better presented as %input.

As requested by the reviewer, we have changed the ChIP data in Figure 5f and h as % of input.

12) The results in figure 6 are very convincing and it is clear that the lack of HIF1a in B cells has an effect on disease pathogenesis. However, this does not reflect the rather limited effect that it has on IL-10 production, suggesting that Bregs may suppress via other mechanisms. The authors should checked at least all the IL-10 independent mediated mechanisms previously reported in the literature.

We thank the reviewer for these valuable comments. Ray A *et al.* (Int Immunol. 2015 Oct;27(10):531-6.) suggested that Breg cells express suppressive molecules such as ICAM1, CD73, GITRL, FasL and PD-L1, which promote Treg cells and inhibit pathogenic T cell functions. Therefore, we have analyzed the expression of these molecules by FACS in B cells in CIA and EAE models. As shown in supplementary figures 6b-c and 7b-c, there was no difference in the expression of these surface markers between WT and *Hif1a*-deficient B cells in CIA and EAE models. In addition, we also measured the levels of total IgGs and additional T cell suppressive cytokines such as TGF β and IL-35 in the serum of WT and *Hif1a*-deficient mice after CII or MOG immunization. We could not observe any significant difference in IgGs

or IL-35 and TGF β levels between WT and *Hif1a*-deficient mice (Supplementary Figure 6a, 6g; Figure 7d, 7e and Supplementary Figure 7a).

13) Figure 8 shows the suppressive capacity of Hif-1a-deficient B cells are rescued by overexpression of IL10. This system is very artificial and it is likely that forced overexpression of IL10 in any B cell subset would result in a suppressive phenotype.

The reviewer is right. However, it is reported that IL-10-producing B cells may not be a specific B cells lineage and that B cells could gain the capacity of IL-10 production in inflammatory microenvironment (Rosser EC et al, Nat Med. 2014;20:1334-9). Moreover, there was no commonly proven transcriptional marker for isolating IL-10-producing B cells. Since IL-10-producing B cells were enriched in CD1d^{hi}CD5⁺ population (Tedder TF, J Immunol 2015; 194:1395-1401) and we found a significantly impaired IL-10 production by CD1d^{hi}CD5⁺ B cells (Figure 3d,e), we sorted *Hif1a*-deficient CD1d^{hi}CD5⁺ population for our rescue experiment.

Reviewer #3

This is a well-conducted study that provides strong evidence, using a variety of strategies that HIF-1a contributes to, but is not essential, for IL-10 production by B cells. While all B cell subsets can produce IL-10, they don't all regulate the severity of autoimmune disease in mice. Thus the study focuses on HIF-1a in CD1dhiCD5+ B cells. However, IL-10 production in CD1dhiCD5+ HIF-1a-deficient B cells was not shown. A reduction in IL-10 production is critical to the interpretation of the data shown. The authors blur the lines between IL-10 production by B cells and regulation of inflammation by B cells producing IL-10. B cells do not switch into an IL-10-producing regulatory B cell subset. They are regulatory because they have the capacity to produce high levels of IL-10. The two are not synonymous. A connection between HIF-1a and IL-10 production in macrophages has already been reported. In addition, a role for HIF-1a in metabolism in macrophages and Tr1 cells has also been reported. These studies should have been discussed and thus reduce the novelty of the current study.

1. The article incorrectly assumes that B cells differentiate or switch into IL-10 producing cells (page 4, line 56). All B cell subsets can produce IL-10 given the appropriate stimuli. There is no "switching" into a regulatory pathway. Thus there is no "official" IL-10-producing regulatory B cell subset as inferred in this manuscript. The authors are encouraged to read the following review for clarification: PMID: 25663677.

As suggested by the reviewer, we have revised the wording of IL-10-producing regulatory B cells in our manuscript and we have been precise in the populations described. The review has also been included in the manuscript.

2. For all figures, please provide the number of mice or data points represent in the line and bar graphs. The number of independent experiments is not sufficient.

Please note that we have increased sample sizes during the revision process. We now present numbers of data points and mice for each figure in the figure legends.

3. How do the authors explain that there is a reduction in T2-MZP, but not a subsequent reduction in MZ B cells? T2-MZP are not regulatory B cells. They are the precursor to MZ B cells. Likewise, CD1dhiCD5+ B cells are a subset of PC B cells and not a separate lineage.

The reviewer is right. In the previous Figure 2c, MZ B cells were only gated by CD19, CD21 and CD23 staining. Therefore, to further clarify MZ B cells, we have performed a specific gating strategy. We, now, used CD19, CD21, CD23, IgM, IgD and CD1d staining to measure MZ B cells in WT and HIF-1 α deficient mice. As shown in the new figure 2d, MZ B cells

(CD19⁺CD21^{hi}CD23^{lgM^{hi}IgD^{lo}CD1d^{hi}) were decreased in *Hif1a*-deficient mice compared to WT control mice.}

4. Thus there is no defect in regulatory B cell proliferation as stated on lines 154, 163. All the data show in Fig. 3d is that the two cell populations in the steady state have a reduction in the number of cells cycling. Hif1a is known to regulate cell proliferation, so its deletion is consistent with a reduction in cell proliferation, which has nothing to do with a regulatory B cell phenotype or their specific regulation by Hif1a. It would be interesting to know whether the results in Fig. 4 are specific to CD1d^{hi}CD5⁺ B cells or whether similar results would be obtained in other B cells subsets. Without that information, nothing can be concluded about the specificity of Hif1a for CD1d^{hi}CD5⁺ B cells.

The reviewer is right, there were previous studies showing HIF-1a as an important factor for cell proliferation, In order to determine whether the decreased in CD1d^{hi}CD5⁺ B cells and T2-MZP proliferation was specific of these populations, we have performed several additional experiments:

- 1) Proliferation of total B cell by BrdU incorporation reveals no difference in WT and HIF-1 α deficient cells (Figure 3c).
- 2) We have measured the glucose import activity by CD1d^{lo}CD5⁻ B cells from WT and HIF-1a deficient mice. No significant difference could be detected for CD1d^{lo}CD5⁻ B cells ($\Delta Hif1a$) compared to CD1d^{lo}CD5⁻ B cells (WT) (Supplementary figure 4b).
- 3) In addition, we have quantified HIF-1 α protein levels in IL-10⁺ or IL-10⁻ B cells, as well as in CD1d^{hi}CD5⁺ or CD1d^{lo}CD5⁻ B cells. Indeed, IL-10⁺ B cells showed a high level of HIF-1 α protein compared to IL-10⁻ B cells, and high levels of HIF-1 α protein could also be detected in CD1d^{hi}CD5⁺ when compared to CD1d^{lo}CD5⁻ B cells (Supplementary figures 3a and 4a)

Altogether, these new data suggested that the low proliferation state is dependent of HIF-1 α expression in CD1d^{hi}CD5⁺ B cells and that high expression of HIF-1 α correlated with IL-10 production by CD1d^{hi}CD5⁺ B cells.

5. It is unclear why some experiments are performed using total B cells and others are performed using specific B cell subsets. This makes the data difficult to interpret in regards to B cell regulation by HIF-1a.

The reviewer is right, using only sorted CD1d^{hi}CD5⁺ B cells would have been ideal, but this population is only 1-2% of total splenocytes. Hence some experiments such as Western blotting and ChIP could not be performed with sorted CD1d^{hi}CD5⁺ B cells. Nonetheless, we have now more focused on CD1d^{hi}CD5⁺ B cells. We have moved the T2-MZP data into

supplementary information and performed additional experiment using sorted CD1d^{hi}CD5⁺ B cells to analyze HIF-1 α and IL-10 levels.

6. In Fig. 8a, the phenotype of the CD4+IL-10+ T cells should be identified. Certain B cell subsets can support iTreg (CD4+Foxp3+) generation. This could explain much of the data in Fig. 8. IL-10+CD4+ is also a phenotype consistent with Foxp3+ T regulatory cells.

As suggested by the reviewer, in addition of the analyses of CD4⁺IL-10⁺ T cells (Tr1- a subset of Treg population), we have analyzed the Foxp3 expression on CD4 T cells in the transwell co-culture system with CD1d^{hi}CD5⁺ (WT) or CD1d^{hi}CD5⁺(Δ *Hif1a*) B cells (figures 8c). No differences were detected in Foxp3⁺Treg cells when co-culturing with either CD1d^{hi}CD5⁺ (WT) or CD1d^{hi}CD5⁺(Δ *Hif1a*) B cells. Moreover, we have also checked the IL-10⁺Foxp3⁺ T cells in CIA and EAE disease models. As shown in supplementary figure 6e and 7d, no significant difference was found in Foxp3⁺ Treg cells in spleen or draining lymph nodes. Only the proportion of IL-10⁺Foxp3⁺ T cells slightly decreased in *Hif1a* mutant mice, however, this population is low (0.2%) in spleen and draining lymph nodes. Besides, we could not detect any difference in *Hif1a* mutant mice on the expression of B cells surface molecules (ICAM1, CD73, GITRL, FasL and PD-L1) or cytokine levels (IL-35 and TGF β), which have been reported to promote Foxp3⁺CD4⁺T cells (Supplementary figure 6b-c, 7b-c and figure 6g, 7e). Altogether, these data suggested that *Hif1a*-deficient B cells have no or minor biological influence on the Foxp3⁺CD4⁺ Treg cells.

7. There is very little evidence that IL-10 directly suppresses CD4 T cell function as suggested in Fig. 8, line 276. To make that claim, IL-10 would have to be neutralized in the culture and IL-10R-deficient T cells would need to be used.

As requested by the reviewer, we have performed the IL-10 neutralization in CD1d^{hi}CD5⁺B cells co-cultured with T helper cells. As shown in figure 8a, IFN- γ ⁺Th1 cells and IL-17A⁺Th17 cells were significantly decreased in IL-10 neutralized group, but not in the control antibody group, suggesting that CD4⁺ T cell polarization into Th1 or Th17 cells is influenced by IL-10 production by CD1d^{hi}CD5⁺B cells.

8. Fig. 8. No data is provided showing that the IL-10-CD1dhiCD5+ Hif1a-/- B cells actually produce higher levels of IL-10 as compared to controls.

As suggested by the reviewer, we have quantified the levels of IL-10 in IL-10-CD1d^{hi}CD5⁺ (Δ *Hif1a*) B cells. As shown in supplementary figure 8a, IL-10 production was strongly increased in IL-10-CD1d^{hi}CD5⁺ (Δ *Hif1a*) B cells when compared to mock CD1d^{hi}CD5⁺ (Δ *Hif1a*) B cells.

9. Line 358, no definitive data was shown that HIF-1a-dependent B cells are crucial for the differentiation and function of Tr1 cells. Tr1 differentiation and function were not studied.

We have revised the sentence.

10. The authors conclude that HIF-1a is crucial in inducing IL-10 production in B cells. Since all B cell subsets produce IL-10 phrasing such as ..”suggesting that HIF-1 is essential for all subsets of IL-10-producing B cells” is not quite accurate and blurs the actual results.

We thank the reviewer for this valuable comment; the wording has been changed as recommended by the reviewer.

11. It was never demonstrated that IL-10 production in CD1d^{hi}CD5⁺ Hif1a^{-/-} B cells is reduced as compared to controls.

The reviewer is right. Therefore, IL-10 expression and production was analyzed in stimulated sorted CD1d^{hi}CD5⁺ B cells. As shown in figure 3d,e and Supplementary figure 3d, impaired IL-10 expression and production was observed in CD1d^{hi}CD5⁺ Hif1a^{-/-} B cells compared to CD1d^{hi}CD5⁺ WT B cells.

Minor comments

12. Line 54, 82 and 328, encephalomyelitis and experimental autoimmune encephalomyelitis should be “experimental autoimmune encephalomyelitis”.

We apologize for this mistake and have revised it.

13. Journal names in the reference list were not correctly abbreviated.

The reference list has been verified.

14. The sentence ending in line 315 needs to be referenced.

The reference has been added.

15. Line 326, no evidence was provided that the IL-17 producing cells are pathogenic. Many of the experiments were performed using splenocytes, that contain a number of different cell types that can contribute to IL-10 production.

In order to clarify this point, IL-23R⁺ or GM-CSF⁺ Th17 cells known to have pathogenic properties in inflammatory disease (Meyer zu Horste G et al, Cell Rep. 2016; 16: 392–404; El-Behi M. et al, Nat Immunol. 2011;12: 568–575.) were measured in CIA or EAE diseased mice. Increased IL-23R⁺IL-17A⁺ and GM-CSF⁺ IL-17A⁺ cells were observed in HIF-1 α deficient

mice (Figure 6j and 7i). Moreover, as requested by the reviewer, we have measured IL-10 production by enriched splenic B cells in CIA or EAE model. As shown in figures 6g and 7e, the level of IL-10 was reduced in *Hif1a*-deficient splenic B cells, similarly than in total splenocytes (figures 6f and 7d)

16. There are English syntax errors that need to be corrected.

The text has been revised.

Reviewers' comments (30.11.2017):

Reviewer #1 (Remarks to the Author):

In the revised manuscript the authors adequately responded to issues raised by reviewers.

I suggest revision of the title, since it does not clearly specify whether the role of HIF-1alpha in B cells is to mediate or prevent autoimmunity. Furthermore, conditional knockouts are not relevant models for human disease (much less an entire class of human diseases) since there are no naturally occurring circumstances under which loss of expression of a single gene in a single cell type occurs.

Reviewer #2 (Remarks to the Author):

The authors have addressed all the reviewers comments well, apart from the following two relatively minor points from us:

5) On page 6, the authors state "Since HIF-1 α induction by LPS has been already reported to be dependent on NF- κ B signaling¹⁸". However, the reference they cite refers to a study in human monocytes. What evidence do the authors have that the signalling pathways utilised are the same in these different cells/species?

As requested by the reviewer, we have performed lentivirus transfection to knock-down RelA gene, an essential protein for NF- κ B signaling, in LPS stimulated B cells. As shown in supplementary figure 1d, HIF-1 α protein level was decreased when RelA was knocked down by siRNA, suggesting that the NF- κ B signaling is also important in the control of HIF-1a protein levels.

This should state "As shown in supplementary figure 1a"

10) Figures 5c & d. 5c . While other regions of the IL10 promoter may exist that contain high levels of H3K4me3, I still find it surprising that the authors could not this detect at regions more proximal to the TSS (i.e. between -401 and -1067), while they could detect it at more distal regions (i.e. between -2851 and -3354), since my understanding is that H3K4me3 mainly marks gene transcription start sites. Also, it seems that these ChIP experiments do not contain IgG negative controls as the other experiments do e.g. figure 1a. Could the authors please show the data or explain why they are omitting this control?

Reviewer #3 (Remarks to the Author):

The authors responded well to the reviewer comments and the revised manuscript needs no further changes.

Response to Reviewers (12.12.2017)

Re: NCOMMS-17-12195

Reviewer #2 (Remarks to the Author):

The authors have addressed all the reviewer's comments well, apart from the following two relatively minor points from us:

5) On page 6, the authors state "Since HIF-1 α induction by LPS has been already reported to be dependent on NF- κ B signaling¹⁸". However, the reference they cite refers to a study in human monocytes. What evidence do the authors have that the signalling pathways utilised are the same in these different cells/species?

As requested by the reviewer, we have performed lentivirus transfection to knock-down RelA gene, an essential protein for NF- κ B signaling, in LPS stimulated B cells. As shown in supplementary figure 1d, HIF-1 α protein level was decreased when RelA was knocked down by siRNA, suggesting that the NF- κ B signaling is also important in the control of HIF-1 α protein levels.

This should state "As shown in supplementary figure 1a"

Sorry for the misunderstanding, we made an error in the point-by-point response letter, the figure was correctly mentioned in the manuscript.

10) Figures 5c & d. 5c . While other regions of the IL10 promoter may exist that contain high levels of H3K4me3, I still find it surprising that the authors could not this detect at regions more proximal to the TSS (i.e. between -401 and -1067), while they could detect it at more distal regions (i.e. between -2851 and -3354), since my understanding is that H3K4me3 mainly marks gene transcription start sites. Also, it seems that these ChIP experiments do not contain IgG negative controls as the other experiments do e.g. figure 1a. Could the authors please show the data or explain why they are omitting this control?

We agree with the reviewer, H3K4me3 modifications usually occur at transcription start sites. Recently it was reported that the proximal broad H3K4me3 domain is associated with high transcription activity and is highly dynamic in different cell-types and conditions (Xiaoyu Liu et al, Nature 537, 558–562 (2016); Julia Engelhorn et al, Epigenomes 2017, 1(2), 8; Fan Cao et al, Scientific Reports 7, 2186 (2017)). Therefore, we think that the high enrichment of H3K4me3 in HRE I or HRE II regions is probably related to the hypoxic condition used in our experiments. We have discussed this issue in the discussion paragraph. Besides, we included IgG control groups for every ChIP assay (Supplementary Fig 5b, e), where no specific signal is detected when using IgG control antibody.